# From Biased to Unbiased Dynamics: An Infinitesimal Generator Approach

**Timothée Devergne**
CSML & ATSIM, Istituto Italiano di Tecnologia
`timothee.devergne@iit.it`

**Vladimir R. Kostic**
CSML, Istituto Italiano di Tecnologia
University of Novi Sad
`vladimir.kostic@iit.it`

**Michele Parrinello**
ATSIM, Istituto Italiano di Tecnologia
`michele.parrinello@iit.it`

**Massimiliano Pontil**
CSML, Istituto Italiano di Tecnologia
AI Centre, University College London
`massimiliano.pontil@iit.it`

## Abstract

We investigate learning the eigenfunctions of evolution operators for time-reversal invariant stochastic processes, a prime example being the Langevin equation used in molecular dynamics. Many physical or chemical processes described by this equation involve transitions between metastable states separated by high potential barriers that can hardly be crossed during a simulation. To overcome this bottleneck, data are collected via biased simulations that explore the state space more rapidly. We propose a framework for learning from biased simulations rooted in the infinitesimal generator of the process and the associated resolvent operator. We contrast our approach to more common ones based on the transfer operator, showing that it can provably learn the spectral properties of the unbiased system from biased data. In experiments, we highlight the advantages of our method over transfer operator approaches and recent developments based on generator learning, demonstrating its effectiveness in estimating eigenfunctions and eigenvalues. Importantly, we show that even with datasets containing only a few relevant transitions due to sub-optimal biasing, our approach recovers relevant information about the transition mechanism.

## 1 Introduction

Dynamical systems and stochastic differential equations (SDEs) provide a general mathematical framework to study natural phenomena, with broad applications in science and engineering. Langevin SDEs, the main focus of this paper, are widely used to simulate physical processes such as protein folding or catalytic reactions [see e.g. 47, and references therein]. A main objective is to describe the dynamics of the process, forecast its evolution from a starting state, ultimately gaining insights on macroscopic properties of the system.

In molecular dynamics, the motion of a molecule is sampled according to a potential energy $U(x)$, where the state vector $x$ represents the positions of all the atoms. Specifically, the Langevin equation $dX_t = -\nabla U(X_t)dt + \sigma dW_t$ describes the stochastic behavior of the system at thermal equilibrium, where $X_t$ is the random position of the state at time $t$, the scalar $\sigma$ is a multiple of the square root of the system's temperature, and $W_t$ is a vector random variable describing thermal fluctuations (Brownian motion). Most often, the atoms evolve in metastable states that are separated by barriers which can hardly be crossed during a simulation. For instance, for a protein the free energy barrier between the folded and unfolded states is larger than thermal agitation, making the transition between the

two states a rare event. Consequently, long trajectories need to be simulated before such interesting events are observed. In fact, one needs to observe many events to get the relevant thermodynamics (free energy) and kinetics (transition rates) information [34]. Beyond molecular dynamics, the slow mixing behavior of many systems modeled by SDEs is a major bottleneck in the study of rare events, and so designing methodologies which can accelerate the process is paramount.

A general idea to overcome the above problem is to perturb the system dynamics. One important approach which has been put in place in molecular dynamics is the so-called "bias potential enhanced sampling" [25, 44, 11]. The main idea is to add to the potential energy a bias potential $V$, thereby lowering the barrier and allowing the system state to be explored more rapidly. To make this approach tractable in large systems, $V$ is often chosen as a function of a few wisely selected variables called collective variables (CVs). For instance, if a chemical reaction involves a bond breaking, physical intuition suggests to choose the distance between the reactive atoms [26, 29]. However, for complex processes, hand-crafted CVs might be "suboptimal", meaning that some of the degrees of freedom important for the transition are not taken into account, making the biasing process inefficient.

In recent years, machine learning approaches have been employed to find the most relevant CVs [8, 42, 14, 9, 10, 4, 27]. A key idea is to use available dynamical information to construct the CVs [41, 31, 7, 42]. For instance, if one can identify the slowest degrees of freedom of the system, one can accurately describe the transitions between metastable states. These approaches are based on learning the transfer operator of the system, which models the conditional expectation of a function (or observable) of the state at a future time, given knowledge of the state at the initial time. It is learned from the behavior of dynamical correlation functions at large lag times which reflects the slow modes of the system. The leading eigenfunctions of the learned transfer operator can then be used as CVs in biased simulations. Moreover, they provide valuable insights into the transition mechanism, such as the location of the transition state ensemble [48]. Still, this approach suffers from the same shortcoming described above, namely if the system is slowly mixing, long trajectories are needed to learn the transfer operator and extract good eigenfunctions.

More recently, there has been growing interest in learning the infinitesimal generator of the process [15, 1, 50, 20], which allows one to overcome the difficult choice of the lag-time. The statistical learning properties of generator learning have been addressed in [21], where an approach based on the resolvent operator has been proposed in order to bypass the unbounded nature of the generator. However the key difficulty of learning from biased simulations remains an open question. In this work, we prove that the infinitesimal generator is the adequate tool to deal with dynamical information from biased data. Leveraging on the statistical learning considerations in [23, 21], we introduce a novel procedure to compute the leading eigenpairs of the infinitesimal generator from biased dynamics, opening the doors to numerous applications in computational chemistry and beyond.

**Contributions** In summary, our main contributions are: **1)** We introduce a principle approach, based on the resolvent of the generator, to extract dynamical properties from biased data; **2)** We present a method to learn the generator from a prescribed dictionary of functions; **3)** We introduce a neural network loss function for learning the dictionary, with provable learning guarantees; **4)** We report experiments on popular molecular dynamics benchmarks, showing that our approach outperforms state-of-the-art transfer operator and recent generator learning approaches in biased simulations. Remarkably, even with datasets containing only a few relevant transitions due to sub-optimal biasing, our method effectively recovers crucial information about the transition mechanism.

**Paper organization** In Section 2, we introduce the learning problem. Section 3 explores limitations of transfer operator approaches. In Section 4, we review a recent generator learning approach [21] and adapt it to nonlinear regression with a finite dictionary of functions. Section 5 presents our method for learning from biased dynamics. Finally, in Section 6, we report our experimental findings.

## 2   Learning dynamical systems from data

In this section, we address learning stochastic dynamical systems from data. After introducing the main objects, we review existing data-driven approaches and conclude with practical challenges. We ground the discussion in the recently developed statistical learning theory, [22–24], contributing in particular to the existence of physical priors and feasibility of data acquisition for successful learning.

**Stochastic differential equations (SDEs) and evolution operators** While our observations in the paper naturally extend to general forms of SDEs [see e.g. 33], to simplify the exposition, we focus on the Langevin equation, which is most relevant to our discussion of biased simulations. Specifically, we consider the overdamped Langevin equation

$$dX_t = -\nabla U(X_t)dt + \sqrt{2\beta^{-1}}dW_t \quad \text{and} \quad X_0 = x, \tag{1}$$

describing dynamics in a (state) space $\mathcal{X} \subseteq \mathbb{R}^d$, governed by the *potential* $V : \mathbb{R}^d \to \mathbb{R}$ at the *temperature* $\beta^{-1} = k_B T$, where $W_t$ is a $\mathbb{R}^d$-dimensional standard Brownian motion.

The SDE (1) admits a unique strong solution $X = (X_t)_{\geq 0}$ that is a Markov process to which we can associate the semigroup of Markov *transfer operators* $(\mathcal{T}_t)_{t \geq 0}$ defined, for every $t \geq 0$, as

$$[\mathcal{T}_t f](x) := \mathbb{E}[f(X_t)|X_0 = x], \quad x \in \mathcal{X}, f \colon \mathcal{X} \to \mathbb{R}. \tag{2}$$

For (1) the distribution of $X_t$ converges to the *invariant measure* $\pi$ on $\mathcal{X}$ called the Boltzmann distribution, given by $\pi(dx) \propto e^{-\beta V(x)}dx$. In such cases, one can define the semigroup on $L_\pi^2(\mathcal{X})$, and characterize the process by the *infinitesimal generator*

$$\mathcal{L} := \lim_{t \to 0^+}(\mathcal{T}_t - I)/t$$

defined on the Sobolev space $H_\pi^{1,2}(\mathcal{X})$ of functions in $L_\pi^2(\mathcal{X})$ whose gradient are also in $L_\pi^2(\mathcal{X})$. The transfer operator and the generator are linked one to another by the formula $\mathcal{T}_t = \exp(t\mathcal{L})$. Moreover, it can be shown (see Appendix A) that the generator $\mathcal{L}$ acts on $f \colon \mathcal{X} \to \mathbb{R}$ as

$$\mathcal{L}f = -\langle \nabla U, \nabla f \rangle + \beta^{-1}\Delta f, \tag{3}$$

which, integrating by parts, gives $\int(\mathcal{L}f)g\,d\pi = -\beta^{-1}\int\langle \nabla f, \nabla g \rangle\,d\pi = \int f(\mathcal{L}g)d\pi$, showing that $\mathcal{L}$ is self-adjoint. If $\mathcal{L}$ has only a discrete spectrum, one can solve (1) by computing the spectral decomposition

$$\mathcal{L} = \sum_{i \in \mathbb{N}}\lambda_i f_i \otimes f_i, \tag{4}$$

Using (2) and the exponential relation between the transfer operator and the generator, one can write

$$[\mathcal{T}_t f](x) := \mathbb{E}[f(X_t)|X_0 = x] = \sum_{i \in \mathbb{N}}e^{t\lambda_i}f_i(x)\langle f_i, f \rangle, \quad x \in \mathcal{X}, f \colon \mathcal{X} \to \mathbb{R} \tag{5}$$

where the timescales of the process appear as the inverses of the generator eigenvalues. Consequently, the eigenpairs of the generator offer valuable insight about the transitions within the studied system.

**Learning from simulations** The main difference underpinning the development of learning algorithms for the *transfer operator* and the *generator* lies in the nature of the data used. While for the transfer operator we can *only* observe a *noisy* evaluation of the output to learn a compact operator, in the case of the generator, knowing the drift and diffusion coefficients allows us to compute the output, albeit at the cost of learning an *unbounded differential operator*. Consequently, learning methods for the former align with vector-valued regression in function spaces [22], whereas methods for the latter, as discussed in the following section, are more akin to physics-informed regression algorithms. In both settings, we learn operators defined on a function (hypothesis) space, formed by the linear combinations of a prescribed set of basis functions (dictionary) $z_j : \mathcal{X} \to \mathbb{R}, j \in [m]$,

$$\mathcal{H} := \left\{ h_u = \sum_{j \in [m]}u_j z_j \,\middle|\, u = (u_1, \ldots, u_m) \in \mathbb{R}^m \right\}. \tag{6}$$

The choice of the dictionary, instrumental in designing successful learning algorithms, may be based on prior knowledge on the process or learned from data [24, 30, 50]. The space $\mathcal{H}$ is naturally equipped with the geometry induced by the norm $\|h_u\|_{\mathcal{H}}^2 := \sum_{j=1}^m u_j^2$. Moreover, every operator $A \colon \mathcal{H} \to \mathcal{H}$ can be identified with matrix $\mathsf{A} \in \mathbb{R}^{m \times m}$ by $Ah_u = z(\cdot)^\intercal \mathsf{A}u$. In the following, we will refer to $A$ and $\mathsf{A}$ as the same object, explicitly stating the difference when necessary.

**Transfer operator learning** Learning the transfer operator $\mathcal{T}_t$ can be simply seen as the vector-valued regression problem [22], in which the action of $\mathcal{T}_t \colon L_\pi^2(\mathcal{X}) \to L_\pi^2(\mathcal{X})$ on the domain $\mathcal{H} \subseteq L_\pi^2(\mathcal{X})$ is estimated by an operator $\widehat{T}_t \colon \mathcal{H} \to \mathcal{H}$. This aims to minimize the mean square error (MSE) w.r.t. the invariant distribution. Given a dataset $\mathcal{D}_n := (x_i, y_i = x_{i+1})_{i=1}^n$ of time-lag $t > 0$ consecutive states from a trajectory of the process, a common approach is to minimize the regularized empirical MSE, leading to the ridge regression (RR) estimator $\widehat{T}_\gamma := \widehat{\mathsf{C}}_\gamma^{-1}\widehat{\mathsf{C}}_t$, where the empirical covariance matrices are $\widehat{\mathsf{C}} = \frac{1}{n}\sum_{i \in [n]}z(x_i)z(x_i)^\intercal$ and $\widehat{\mathsf{C}}_t = \frac{1}{n}\sum_{i \in [n]}z(x_i)z(y_i)^\intercal$. We then estimate the eigenpairs $(\lambda_i, f_i)$ in (4) by the eigenpairs $(\widehat{\mu}_i, \widehat{u}_i)$ of $\widehat{T}_\gamma$ as $\widehat{\lambda}_i := \ln(\widehat{\mu}_i/t)$ and $\widehat{f}_i := z(\cdot)^\intercal \widehat{u}_i$.

We stress that transfer operator approaches crucially relies on the definition of the time-lag $t$ from which dynamics is observed. Setting this value is a delicate task, depending on the events one wants to study. If $t$ is chosen too small, the cross-covariance matrices will be too noisy for slowly mixing processes. On the other hand, if $t$ is too large, because the relevant phenomena occur at large time scales, a very long simulation is needed to compute the covariance matrices. In order to overcome this problem biased simulations can be used, which we discuss next.

## 3 Learning from biased simulations

As discussed above, in molecular dynamics, the desired physical phenomena often cannot be observed within an affordable simulation time. To address this, one solution is to modify the potential,

$$U'(x) := U(x) + V(x), \quad x \in \mathcal{X}$$

where we assume that the introduced perturbation (a form of bias in the data) $V(x)$ is known. For example the bias potential $V$ may be constructed from previous system states to promote transitions to not yet visited regions. One of the prototypical examples is metadynamics [25], where $V$ is a sum of Gaussians built on the fly in order to reduce the barrier between metastable states. However, the bias potential alters the invariant distribution [12], making it challenging to recover the unbiased dynamics from biased data. Denoting the invariant measure of the perturbed process by $\pi'$ and its generator by $\mathcal{L}' \colon H^{1,2}_{\pi'}(\mathcal{X}) \to H^{1,2}_{\pi'}(\mathcal{X})$, our principal objective is thus to:

*Gather data from simulations generated by $\mathcal{L}'$ to learn the spectral decomposition of the unperturbed generator $\mathcal{L}$.*

To tackle this problem, we note that since the eigenfunctions of the generator $\mathcal{L}$ are also eigenfunctions of every transfer operator $\mathcal{T}_t = e^{t\mathcal{L}}$, we can address the related problem of learning the transfer operator from perturbed dynamics. Unfortunately, there is an inherent difficulty in doing so. While one *typically knows the perturbation* in the generator, that is $\mathcal{L}' = \mathcal{L} + \langle \nabla V, \nabla(\cdot) \rangle$, this knowledge is not easily transferred to the perturbation of the transfer operator. Indeed, recalling that $\mathcal{T} := \mathcal{T}_1 = e^{\mathcal{L}}$, and since the differential operator $\langle \nabla V, \nabla(\cdot) \rangle$ in general does not share the same eigenstructure of $\mathcal{L}$, one has that

$$\mathcal{T}' := e^{\mathcal{L}'} = e^{\mathcal{L} - \langle \nabla V, \nabla(\cdot) \rangle} \neq \mathcal{T} e^{-\langle \nabla V, \nabla(\cdot) \rangle}.$$

Simply put, the generator depends linearly on the bias, while the transfer operator does not. One strategy to overcome the data distribution change, is to *adapt* the notion of the risk. To discuss this idea, recall that the invariant distribution of overdamped Langevin dynamics is the Boltzmann distribution defined by the potential. Hence, we have that

$$\pi(dx) = \frac{e^{-\beta U(x)}dx}{\int e^{-\beta U(x)}dx}, \ \pi'(dx) = \frac{e^{-\beta U'(x)}dx}{\int e^{-\beta U'(x)}dx} \ \text{ and } \ \frac{d\pi}{d\pi'}(x) = \frac{e^{\beta V(x)}}{\int e^{\beta V(x)}\pi'(dx)} \tag{7}$$

where the last term is the Radon-Nikodym derivative, which exposes the data-distribution change. Consequently, we can express the covariance operators for the unperturbed process as weighted expectations of the perturbed data features

$$\mathbf{C} = \mathbb{E}_{X' \sim \pi'} \left[ \tfrac{d\pi}{d\pi'}(X')z(X')z(X')^\mathsf{T} \right]. \tag{8}$$

However, since the transition kernel of the process $(X'_t)_{t \geq 0}$ generated by $\mathcal{L}'$ is different from that of the original process, the above reasoning does not hold for the cross-covariance matrix, that is,

$$\mathbf{C}_t := \mathbb{E}_{X_0 \sim \pi'} \left[ \tfrac{d\pi}{d\pi'}(X_0) \, z(X_0)z(X_t)^\mathsf{T} \right] \neq \mathbb{E}_{X'_0 \sim \pi'} \left[ \tfrac{d\pi}{d\pi'}(X'_0) \, z(X'_0)z(X'_t)^\mathsf{T} \right] =: \mathbf{C}'_t,$$

Consequently, the estimator $\widehat{T}_t$ obtained by minimizing the reweighed risk functional $\mathcal{R}'(\widehat{T}_t) := \mathbb{E}_{X_0 \sim \pi'} \left[ \tfrac{d\pi}{d\pi'}(X_0) \, \|z(X'_t) - \widehat{T}_t^\mathsf{T} z(X_0)\|_2^2 \right]$ *does not* minimize the true risk since $\mathcal{R}'(\widehat{T}_t) \neq \mathcal{R}(\widehat{T}_t)$. Despite this difference, whenever the perturbation is small or controlled and the *time-lag $t$ is small enough*, estimating the true transfer operator of the process from the perturbed dynamics via reweighed covariance/cross-covariance operators has been systematically used as the state-of-the art approach in the field of atomistic simulations [7, 9, 31, 49]. The (limited) success of such approaches is based on a delicate balance of a small enough lag-time and biased potential, since for small $t > 0$ one can approximate $\mathbf{C}_t$ by $\mathbf{C}'_t$ and minimize $\mathcal{R}'(T) \approx \mathcal{R}(T)$ over $T \colon \mathcal{H} \to \mathcal{H}$.

# 4 Infinitesimal generator learning

In this section, we address generator learning. While there has been significant progress on this topic [24, 15, 35, 50, 20], we follow the recent approach in [21] for learning the generator $\mathcal{L}$ on an a priori fixed hypothesis space $\mathcal{H}$ through its resolvent. Leveraging on its strong statistically guarantees, we adapt it from kernel regression to nonlinear regression over a dictionary of basis functions, setting the stage for the development of our deep-learning method.

While transfer operator learning does not require any prior knowledge of the system's drift and diffusion, making use of this information helps learning the generator and avoids the need for setting the time lag parameter. We briefly discuss how to achieve this for over-damped Langevin processes when the constant diffusion term is known. We estimate the generator *indirectly* via its resolvent $(\eta I - \mathcal{L})^{-1}$, where $\eta > 0$ is a prescribed parameter. To this end, we observe that the action of the resolvent in $\mathcal{H}$ can be expressed as $((\eta I - \mathcal{L})^{-1} h_u)(x) = \chi_\eta(x)^\top u$, where $\chi_\eta$ is the embedding of the resolvent $(\eta I - \mathcal{L})^{-1}$ into $\mathcal{H}$, given by $\chi_\eta(x) = \int_0^\infty \mathbb{E}[z(X_t) e^{-\eta t} \mid X_0 = x] dt$, $x \in \mathcal{X}$, see [21]. We then aim to approximate $\chi_\eta(x) \approx \mathrm{G}^* z(x)$ by a matrix $\mathrm{G} \in \mathbb{R}^{m \times m}$. Unfortunately the embedding of the resolvent is not known in close form. To overcome this, we contrast the resolvent by defining a *regularized energy kernel* $\mathfrak{E}_\pi^\eta \colon H_\pi^{1,2}(\mathcal{X}) \times H_\pi^{1,2}(\mathcal{X}) \to \mathbb{R}$, given by $\mathfrak{E}_\pi^\eta[f,g] = \mathbb{E}_{x \sim \pi}[\eta f(x) g(x) - f(x)[\mathcal{L}g](x)]$, which using (3) becomes

$$\mathfrak{E}_\pi^\eta[f,g] = \mathbb{E}_{x \sim \pi}\left[\eta f(x) g(x) + f(x) \nabla U(x)^\top \nabla g(x) - \tfrac{1}{\beta} f(x) \Delta g(x)\right], \tag{9}$$

and, due to the identity $\int f \mathcal{L} g \, d\pi = -\beta^{-1} \int (\nabla f)^\top (\nabla g) d\pi$, also

$$\mathfrak{E}_\pi^\eta[f,g] = \mathbb{E}_{x \sim \pi}\left[\eta f(x) g(x) + \tfrac{1}{\beta} \sum_{k \in [d]} \partial_k f(x) \partial_k g(x)\right]. \tag{10}$$

Since $\mathcal{L}$ is negative semi-definite, the above kernel induces the *regularized squared energy norm* $\mathfrak{E}_\pi^\eta \colon H_\pi^{1,2}(\mathcal{X}) \to [0, +\infty)$ by $\mathfrak{E}_\pi^\eta[f] := \mathfrak{E}_\pi^\eta[f,f] = \mathbb{E}_{x \sim \pi}[\eta f^2(x) - f(x)[\mathcal{L}f](x)]$. It counteracts the resolvent and balances the transient dynamics (energy) of the process with the invariant distribution $\pi$. In a nutshell, instead of using the mean square error of $f(x) := \|\chi_\eta(x) - \mathrm{G}^\top z(x)\|_2$ to define the risk, we "*fight fire with fire*" and penalize the energy to formulate the *generator regression problem*

$$\min_{G \colon \mathcal{H} \to \mathcal{H}} \mathcal{R}_\partial(G) \equiv \mathcal{R}_\partial(\mathrm{G}) := \mathfrak{E}_\pi^\eta\big[\|\chi_\eta(\cdot) - \widehat{\mathrm{G}}^\top z(\cdot)\|_2\big]. \tag{11}$$

Indeed, this risk overcomes the difficulty of not knowing $\chi_\eta$. To show this, let us define the space $\mathcal{H}_\pi^\eta(\mathcal{X}) := \{f \in H_\pi^{1,2}(\mathcal{X}) \mid \mathfrak{E}_\pi^\eta[f] < \infty\}$ associated to the energy norm $\|f\|_{\mathcal{H}_\pi^\eta} := \sqrt{\mathfrak{E}_\pi^\eta[f]}$, and recalling that the operator $G \colon \mathcal{H} \to \mathcal{H}$ is identified with a matrix $\mathrm{G} \in \mathbb{R}^{m \times m}$ via $G h_u = z(\cdot)^\top (\mathrm{G}u)$, define the (injection) operator $\mathcal{Z} \colon \mathbb{R}^m \to \mathcal{H}_\pi^\eta$ by $\mathcal{Z}u = z(\cdot)^\top u$, for every $u \in \mathbb{R}^m$. Then, since $\mathrm{HS}\,(\mathbb{R}^m, \mathcal{H}_\pi^\eta) \equiv \mathrm{HS}\,(\mathcal{H}, \mathcal{H}_\pi^\eta)$, the norm is the sum of squared $\mathcal{H}_\pi^\eta$ norm over the standard basis in $\mathbb{R}^m$, and one obtains

$$\mathcal{R}_\partial(G) = \|(\eta I - \mathcal{L})^{-1} - G\|_{\mathrm{HS}(\mathcal{H}, \mathcal{H}_\pi^\eta)}^2$$
$$= \underbrace{\|P_\mathcal{H}(\eta I - \mathcal{L})^{-1} - G\|_{\mathrm{HS}(\mathcal{H}, \mathcal{H}_\pi^\eta)}^2}_{\text{projected problem}} + \underbrace{\|(I - P_\mathcal{H})(\eta I - \mathcal{L})^{-1}\|_{\mathrm{HS}(\mathcal{H}, \mathcal{H}_\pi^\eta)}^2}_{\text{representation error } \rho(\mathcal{H})}, \tag{12}$$

where $P_\mathcal{H}$ is the orthogonal projector in $\mathcal{H}_\pi^\eta(\mathcal{X})$ onto $\mathcal{H}$. In learning theory $\rho(\mathcal{H})$ is known as the approximation error of the hypothesis space $\mathcal{H}$ [see e.g. 43]. While this error may vanish for infinite-dimensional spaces, when $\mathcal{H}$ is finite dimensional, controlling $\rho(\mathcal{H})$ is crucial to achieving statistical consistency. This can be accomplished by minimizing (11), which is equivalent to

$$\min_{G \colon \mathcal{H} \to \mathcal{H}} \|P_\mathcal{H}(\eta I - \mathcal{L})^{-1} - G\|_{\mathrm{HS}(\mathcal{H}, \mathcal{H}_\pi^\eta)}^2 = \|\mathcal{Z}(\mathcal{Z}^* \mathcal{Z})^\dagger \mathcal{Z}^*(\eta I - \mathcal{L})^{-1} \mathcal{Z} - \mathcal{Z}\mathrm{G}\|_{\mathrm{HS}(\mathbb{R}^m, \mathcal{H}_\pi^\eta)}^2 \tag{13}$$

where $(\cdot)^\dagger$ is the Moore-Penrose's pseudoinverse. Using the covariance matrices

$$\mathcal{Z}^*(\eta I - \mathcal{L})^{-1} \mathcal{Z} = \mathrm{C} = \big(\mathbb{E}_{x \sim \pi}[z_i(x) z_j(x)]\big)_{i,j \in [m]}, \quad \text{and} \quad \mathrm{W} = \mathcal{Z}^* \mathcal{Z} = \big(\mathfrak{E}_\pi^\eta[z_i, z_j]\big)_{i,j \in [m]}, \tag{14}$$

w.r.t. the invariant distribution and energy, respectively, gives the ridge regularized (RR) solution $\mathrm{G} = (\mathrm{W} + \gamma \mathrm{I})^{-1} \mathrm{C}$, $\gamma > 0$. The induced RR estimator of the resolvent, $G_{\eta,\gamma} \colon \mathcal{H} \to \mathcal{H}$ is given, for every $h_u \in \mathcal{H}$, by $G_{\eta,\gamma} h_u := \mathcal{Z}(\mathrm{W} + \gamma \mathrm{I})^{-1} \mathrm{C}u = z(\cdot)^\top (\mathrm{W} + \gamma \mathrm{I})^{-1} \mathrm{C}u$, and it can be estimated given data from $\pi$ by replacing expectation and the energy in (14) with their empirical counterparts.

# 5 Unbiased learning of the infinitesimal generator from biased simulations

In this section, we present the main contributions of this work: approximating the leading eigenfunctions (corresponding to the slowest time scales) of the infinitesimal generator from biased data. While the general pipeline for the method can be found in figure 1, in the following, we first address regressing the generator on an a priori fixed hypothesis space $\mathcal{H}$. Then we introduce our deep-learning method to either build a suitable space $\mathcal{H}$, or even directly learn the eigenfunctions.

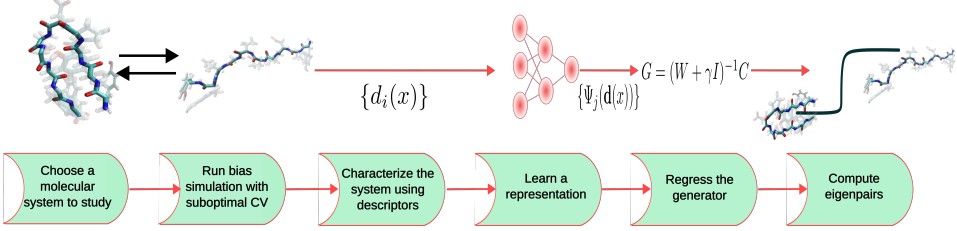

Figure 1: Pipeline of our method: from biased simulations to timescales and metastable states.

**Unbiasing generator regression**   Whenever $\pi$ is absolutely continuous w.r.t. $\pi'$, the regularized energy kernel (9) satisfies the simple identity

$$\mathfrak{E}_\pi^\eta[f,g] = \mathfrak{E}_{\pi'}^\eta\big[f\sqrt{\tfrac{d\pi}{d\pi'}}, g\sqrt{\tfrac{d\pi}{d\pi'}}\big], \quad f,g \in H_\pi^{1,2}(\mathcal{X}), \tag{15}$$

which, recalling the rightmost equation in (7), implies that when the bias $V$ and the diffusion coefficient $\beta$ are known, the energy kernel can be empirically estimated through samples from $\pi'$ via (10). Moreover, when the potential $U$ is known too, we can use (9). Now, leveraging on (15) we directly obtain that

$$\mathcal{R}_\partial(G) \equiv \mathcal{R}_\partial(\mathrm{G}) = \mathfrak{E}_{x\sim\pi'}^\eta\big[\|\chi_\eta(x) - \widehat{\mathrm{G}}^\mathsf{T}z(x)\|_2 \sqrt{\tfrac{d\pi}{d\pi'}(x)}\big] \le \kappa_V \mathfrak{E}_{x\sim\pi'}^\eta \|\chi_\eta(x) - \widehat{\mathrm{G}}^\mathsf{T}z(x)\|_2 \tag{16}$$

where $\kappa_V = \operatorname{ess\,sup}_{x\sim\pi'} \tfrac{d\pi}{d\pi'}(x)$, which recalling (7) is finite whenever the bias $V$ is essentially bounded. Therefore, in sharp contrast to transfer operator learning, whenever the true embedding $\chi_\eta(x)$ can be estimated, one can derive principled estimators of the true generator $\mathcal{L}$'s dominant eigenpairs from the biased dynamics generated by $\mathcal{L}'$. This is established by the following proposition, the proof of which is presented in Appendix B.

**Theorem 1.** *Let $\mathcal{D}_n = (x_i')_{i\in[n]}$ be the biased dataset generated from $\pi'$. Let $w(x) = e^{\beta V(x)}$ and define the empirical covariances w.r.t. the empirical distribution $\widehat{\pi}' = n^{-1}\sum_{i\in[n]}\delta_{x_i'}$ by*

$$\widehat{\mathrm{C}} = \big(\mathbb{E}_{x'\sim\widehat{\pi}'}[w(x')z_i(x')z_j(x')]\big)_{i,j\in[m]} \quad \text{and} \quad \widehat{\mathrm{W}} = \big(\mathfrak{E}_{\widehat{\pi}'}^\eta[\sqrt{w}z_i, \sqrt{w}z_j]\big)_{i,j\in[m]}. \tag{17}$$

*Compute the eigenpairs $(\nu_i, v_i)_{i\in[m]}$ of the RR estimator $\widehat{\mathrm{G}}_{\eta,\gamma} = (\widehat{\mathrm{W}} + \eta\gamma\mathrm{I})^{-1}\widehat{\mathrm{C}}$, and estimate the eigenpairs in (4) as $(\widehat{\lambda}_i, \widehat{f}_i) = (\eta - 1/\nu_i, z(\cdot)^\mathsf{T}v_i)$. If the elements of $\mathcal{H}$ and their gradients are essentially bounded, and $\lim_{m\to\infty}\rho(\mathcal{H}) = 0$, then for every $\varepsilon > 0$, there exist $(m,n,\gamma) \in \mathbb{N}\times\mathbb{N}\times\mathbb{R}_+$, such that, for every $i \in [m]$, $|\lambda_i - \widehat{\lambda}_i| \le \varepsilon$ and $\sin_{L_\pi^2}(\sphericalangle(f_i, \widehat{f}_i)) \le \varepsilon$, with high probability.*

Note that, due to the form of the estimator, the normalizing constant $\int w(x)dx$ does not need be computed. Moreover, relying on the upper bound in (16) we can alternatively compute $\widehat{\mathrm{C}}$ and $\widehat{\mathrm{W}}$ without the weights $w$ and still ensure that the above result holds true.

**Neural network based learning**   Theorem 1 guarantees successful estimation of the eigendecomposition of the generator in (4) whenever the energy-based *representation error* $\rho(\mathcal{H})$ in (12) is controlled. It is therefore natural to minimize $\rho(\mathcal{H})$ by choosing an appropriate basis function $z_i$'s. Inspired by the recent work [24], we parameterize them by a neural network, and optimize them to span the leading invariant subspace of the generator.

Let $z^\theta = (z_i^\theta)_{i\in[m]}: \mathcal{X} \to \mathbb{R}^m$ be a neural network (NN) embedding parameterized by $\theta \in \Theta$ weights with continuously differentiable activation functions, and let $\lambda_i^\theta$, $i \in [m]$, be real non-positive

(trainable) weights. We propose to optimize the NN to find the slowest time-scales $\lambda_i^\theta$ that solve the eigenvalue equation $\mathcal{L}z_i^\theta = \lambda_i^\theta z_i^\theta$, $i \in [m]$. Letting $\mathcal{Z}_\theta \colon \mathbb{R}^m \to \mathcal{H}_\pi^\eta(\mathcal{X})$ be the (parameterized) injection operator, given, for every $u \in \mathbb{R}^m$ by $\mathcal{Z}_\theta u = \sum_{i \in [m]} z_i^\theta u_i$, and denoting $\Lambda_\theta^\eta = (\eta I - \mathrm{diag}(\lambda_1^\theta, \ldots, \lambda_m^\theta))^{-1}$, the eigenvalue equations for the resolvent then become $(\eta I - \mathcal{L})^{-1} \mathcal{Z}_\theta = \mathcal{Z}_\theta \Lambda_\theta^\eta$. In other words, we aim to find the best rank-$m$ decomposition of resolvent $(\eta I - \mathcal{L})^{-1} \approx \mathcal{Z}_\theta \Lambda_\theta^\eta \mathcal{Z}_\theta^*$. Therefore, for some hyperparameter $\alpha \geq 0$ we introduce the loss

$$\mathcal{E}_\alpha(\theta) := \|(\eta I - \mathcal{L})^{-1} - \mathcal{Z}_\theta \Lambda_\eta^\theta \mathcal{Z}_\theta^*\|_{\mathrm{HS}(\mathcal{H}_\pi^\eta)}^2 - \|(\eta I - \mathcal{L})^{-1}\|_{\mathrm{HS}(\mathcal{H}_\pi^\eta)}^2 + \alpha \sum_{i,j \in [m]} (\langle z_i^\theta, z_j^\theta \rangle_{L_\pi^2} - \delta_{i,j})^2.$$

While the first term measures the approximation error in the energy space, it cannot be used as a loss, because the action of the resolvent is not known. To mitigate this, the second term is introduced, under the assumption that $(\eta I - \mathcal{L})^{-1} \in \mathrm{HS}\left(\mathcal{H}_\pi^\eta(\mathcal{X})\right)$ (see Appendix C for a discussion). The third term is optional; specifically, if the goal is not only to identify the proper invariant subspace of the generator ($\alpha = 0$), but also to optimize the neural network to extract eigenfunctions as features, then this last term ($\alpha > 0$) encourages the orthonormality of features in $L_\pi^2(\mathcal{X})$, an idea successfully exploited in machine learning and computational chemistry [see e.g. 24, and references therein].

Recalling (14) and denoting by $\mathrm{C}_\theta$ and $\mathrm{W}_\theta$ the covariance matrices associated to the parameterized features, after some algebra, we obtain that

$$\mathcal{E}_\alpha(\theta) = \mathrm{tr}\left[\mathrm{C}_\theta \Lambda_\theta^\eta \mathrm{W}_\theta \Lambda_\theta^\eta - 2\mathrm{C}_\theta \Lambda_\theta^\eta + \alpha(\mathrm{C}_\theta - \mathrm{I})^2\right]. \tag{18}$$

In turn, this can be estimated from biased data by two independent samples $\widehat{\pi}_1'$ and $\widehat{\pi}_2'$ as

$$\mathcal{E}_\alpha^{\widehat{\pi}_1', \widehat{\pi}_2'}(\theta) = \mathrm{tr}\left[(\widehat{\mathrm{C}}_\theta^1 \Lambda_\theta^\eta \widehat{\mathrm{W}}_\theta^2 \Lambda_\theta^\eta + \widehat{\mathrm{C}}_\theta^2 \Lambda_\theta^\eta \widehat{\mathrm{W}}_\theta^1 \Lambda_\theta^\eta)/2 - \widehat{w}^1 \widehat{\mathrm{C}}_\theta^2 \Lambda_\theta^\eta - \widehat{w}_2 \widehat{\mathrm{C}}_\theta^1 \Lambda_\theta^\eta + \alpha(\widehat{\mathrm{C}}_\theta^1 - \widehat{w}_1 \mathrm{I})(\widehat{\mathrm{C}}_\theta^2 - \widehat{w}_2 \mathrm{I})\right], \tag{19}$$

where $\widehat{\mathrm{C}}_\theta^k$ and $\widehat{\mathrm{W}}_\theta^k$ are the empirical covariances given by (17) for distribution $\widehat{\pi}_k'$, while $\widehat{w}^k = \mathbb{E}_{x' \sim \widehat{\pi}_k' x} w(x')$, $k \in [2]$. Importantly, the computational complexity of the loss (19) is of the order $\mathcal{O}(nm^2d)$, where $d$ is the state dimension and $n$ the sample size, however it can be reduced to $\mathcal{O}(nmd)$ (see Appendix C) allowing its application to learn large dictionaries for high-dimensional problems with big amounts of (biased) data.

The following result, linked to controlling of the representation error as detailed in Theorem 1, provides theoretical guarantees for our approach. The proof and discussion are provided in Appendix B.

**Theorem 2.** *Given a compact operator $(\eta I - \mathcal{L})^{-1}$, $\eta > 0$, if $(z^\theta)_{i \in [m]} \subseteq \mathcal{H}_\pi^\eta(\mathcal{X})$ for all $\theta \in \Theta$, then*

$$\mathbb{E}\left[\mathcal{E}_\alpha^{\widehat{\pi}_1', \widehat{\pi}_2'}(\theta)\right] = \overline{w}^2 \, \mathcal{E}_\alpha(\theta) \geq -\sum_{i \in [m]} \frac{\overline{w}^2}{(\eta - \lambda_i)^2}, \quad \text{for all } \theta \in \Theta, \tag{20}$$

*where $\overline{w} = \mathbb{E}_{x \sim \pi'}[w(x)]$. Moreover, if $\alpha > 0$ and $\lambda_{m+2} < \lambda_{m+1}$, then the equality holds if and only if $(\lambda_i^\theta, z_i^\theta) = (\lambda_i, f_i)$ $\pi$-a.e., up to the ordering of indices and choice of eigenfunction signs for $i \in [m]$.*

This theorem provides a justification for minimizing the loss in (19), which can be achieved by stochastic optimization algorithms, to obtain an approximation of either the leading invariant subspace of the resolvent $(\eta I - \mathcal{L})^{-1}$ (without orthonormality loss, i.e. $\alpha = 0$), on which the estimator in Theorem 1 can be computed, or even the individual eigenpairs ($\alpha > 0$). A pseudocode of our method is provided below. The main advantage of this method is that it exploits the knowledge of the process. namely, if only the bias $V$ and the diffusion coefficient $\beta$ are known, recalling (10), the computation of loss relies just of the gradient of the features. On the other hand, the knowledge of the potential can also be exploited via (9). Finally, even if the neural network features are not perfectly learned, one can still resort to Theorem 1 to compute the approximate eigendecomposition of $\mathcal{L}$.

## 6 Experiments

In this section, we test the method described above on well-established [14, 9, 32, 36] molecular dynamics benchmarks, featuring biased simulations of increasing complexity. We first start by showing the efficiency of our method on a simple one dimensional double well potential. We then proceed to the Muller-Brown potential which is a 2D potential, where this time, sampling is

| Algorithm 1: From biased to unbiased dynamics via infinitesimal generator |
| --- |

1: **Parameters** $\eta > 0$ shift of the generator, $m$ number of wanted eigenfunctions, $K$ number of
   optimization steps, $\gamma > 0$ and $\alpha > 0$ regression and NN hyperparameters
2: **Inputs** Dataset $\mathcal{D}_n = (x_\ell)_{\ell \in [n]}$ gathered from a simulation with bias potential $V$
3: Compute weights $w(x_\ell) = \exp(\beta V(x_\ell))$, $\ell \in [n]$, to be used in line 7
4: **if** the dictionary of function $z$ does not already exist **then**
5:     **Initialization**: randomly initialize neural networks weights of $\Lambda^\theta$ and $(z_i^\theta)_{i \in [m]}$, set $k = 0$
6:     **while** $k < K$ **do**
7:         Compute $\widehat{\mathbf{C}}_\theta^j$ and $\widehat{\mathbf{W}}_\theta^j$, $j = 1, 2$, using (17) for two independent batches $\widehat{\pi}_1'$ and $\widehat{\pi}_2'$
8:         Compute loss $\widehat{\mathcal{E}}^{\widehat{\pi}_1', \widehat{\pi}_2'}(\theta)$ using (19) and backpropagate
9:     **end while**
10: **end if**
11: Compute $\widehat{\mathbf{C}}$ and $\widehat{\mathbf{W}}$ using (17) the datatset $\mathcal{D}_n$
12: Compute the eigenpairs $(\nu_i, v_i)_{i \in [m]}$ of $\widehat{\mathbf{G}}_{\eta, \gamma} = (\widehat{\mathbf{W}} + \eta\gamma\mathbf{I})^{-1}\widehat{\mathbf{C}}$
13: **Output** Estimated eigenpairs of $\mathcal{L}$ are $(\widehat{\lambda}_i, \widehat{f}_i) = (\eta - 1/\nu_i, z^\theta(\cdot)^\mathsf{T} v_i)$, $i \in [m]$

accelerated by a bias potential built on the fly. Finally, we study the conformational landscape of alanine dipeptide. This small molecule is a classical testing ground for rare event methods. To showcase the efficiency of our method we analyse two different sets of data both generated in a metadynamics-like approach and showcase the efficiency of our approach, even with a small number of transitions in the training set. The codes used to train the models can be found in the following repository: https://github.com/DevergneTimothee/GenLearn

**One dimensional double well potential**   We first showcase the efficiency of our method on a simple one dimensional toy model. We sample transitions from $U + V$, where $U$ is a double well potential and $V$ is a bias potential. The results are shown Figure 7 in the appendix, where our method clearly outperforms transfer operator approaches and recovers the true underlying dynamics.

**Muller Brown potential with metadynamics biasing**   Muller Brown is a 2 dimensional potential presenting metastable states often studied in the context of enhanced sampling [19, 37, 14, 9]. It presents two minima, with one of them separated into two sub-basins. We thus expect two relevant eigenpairs: the slowest one corresponding to the transition between the two basins and the second slowest one describing the transition between the two sub-basins. However, at low temperature crossing the barrier occurs rarely. To expedite the rate of transition we use metadynamics and instead of having a predefined bias potential, as in the previous section, the bias is built on the fly using metadynamics [25]. The results of the training procedure are presented in Figure 2. We compare the results with deepTICA and a state of the art generator learning approach in [50]. From this figure, we see that we managed to accurately learn the dynamical behavior of the system despite the fact that the dynamics was performed using a bias potential. As expected, it is clearly outperforming transfer operator approaches. We achieve similar or slightly better results (particularly near the transition state) on the qualitative shape of the eigenfunctions. On the other hand, our method performs better than previous work on generator learning on the estimation of eigenvalues, and is the closest to the ground truth eigenvalues. This is likely to be due to the fact that the method in [50] requires the tuning of hyperparameters in the loss function, while in our case, these coefficients are trainable. It should be noted that here, the eigenfunctions were fitted with well-learned features. However, we present in the appendix results where the features are not perfectly learned, but we still manage to recover the eigenfunctions.

**Alanine dipeptide with OPES biasing**   We next treat a more concrete molecular dynamics example with the study of the conformational change of alanine dipeptide in gas phase. It is a molecule containing 22 atoms, of which 10 are heavy. For the remaining of this study, we will only take into account the positions of the heavy atoms, making it a 30 dimensional system. This molecule has widely been used to test methods in enhanced sampling [9, 50, 42]: it presents a conformational change which is a rare event described by the angles $\phi$ and $\psi$. In the studies made on this system, the angle $\phi$ has been shown to be a good CV: the transition between the two states is very well described, and thus a bias potential can easily be built with this CV. On the other hand, the angle $\psi$ misses most of the transition and is a non optimal CV. We generated biased dataset using a variation of

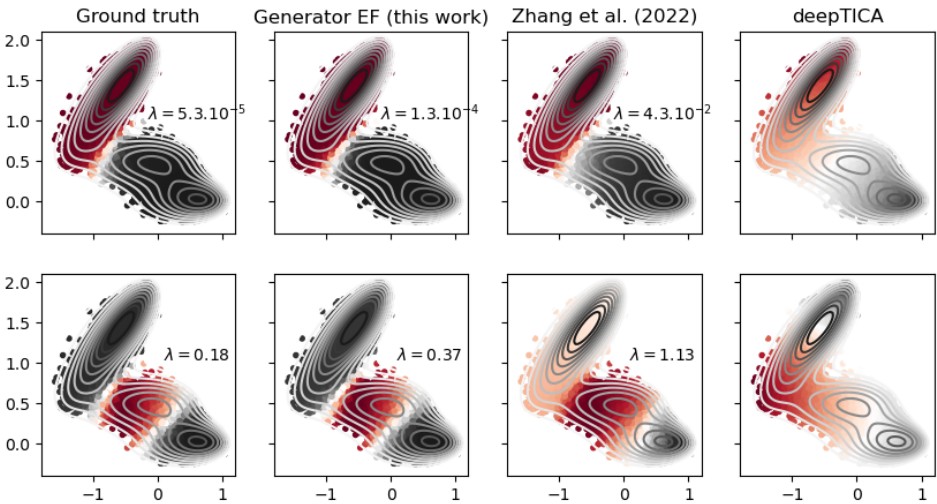

Figure 2: Muller Brown potential. Comparison of the ground truth two first relevant eigenfunctions of the potential (**first column**) with this work (**second column**), transfer operator approach deepTICA [7] (**third column**) and the work of Zhang et al. [50] (**fourth column**). x and y axis are the coordinates of the system and points are colored according to the value of the eigenfunction. The underlying potential is represented by the level lines in white. Associated eigenvalues $\lambda$ are also reported.

metadynamics called on the fly probability enhanced sampling (OPES) [16], which allows a more extensive and faster exploration of the state space than metadynamics [25]:

*Dataset 1:* 800ns simulation, biasing on the $\psi$ dihedral angle, with OPES leading to few transitions between the two states. The bias potential was built during the first 100ns of the simulation. For the remaining 700ns, the potential built during the first part was kept fixed to enhance transitions.
*Dataset 2:* 50ns simulation, biasing on $\phi$ dihedral angle, with OPES leading to many transitions between the two states. The bias potential was built during the first 20ns of the simulation. For the remaining 30ns, the potential built during the first part was kept to enhanced transitions.

Dataset 1 mimics situations where one has only a basic prior knowledge of the system: only a "suboptimal" CV is used yielding to only a few transitions between the metastable states within the affordable simulation time. In order to ensure translational and rotational invariant vectors, we use Kabsch [18] algorithm, which has been used in previous studies [50, 4, 10] to transform the positions of the atoms. The results are presented in Figure 3. Panels **a)** and **b)** display the first and second eigenfunctions learned by our method respectively. Notice that, even though only 2 transitions are present in dataset 1, the first eigenfunction separates the two metastable states, and the second identifies a faster transition in one metastable state. Panel **c)** showcases the good out-of-sample generalization ability of the method. It visualizes the first eigenfunction obtained as above, but this time visualized on points from dataset 2 and in the plane of dihedral angles $\phi$ and $\theta$. Interestingly, we discover that a linear relationship is present in the transition region, in agreement with recent findings in the molecular dynamics literature [6, 19].

To further improve the description of the transition and to enhance the training set without any prior knowledge of the mechanism, one could perform biased simulations using the first eigenfunction. Nonetheless, this is not the scope of this paper. To push our method further and see its capabilities when training on a good dataset, we trained it on Dataset 2. One key quantity in molecular dynamics is the committor function for metastable states A and B, which is defined as the probability of, starting from A, going to B before going back to A. Theory tells us that the committor is linearly related to the first eigenfunction of the generator, a result going back to Kolmogorov [see 5, for a discussion]. This relation is exposed in panel **d)** of Figure 3, when comparing to the committor model obtained in [19] indicating the good performance of our method.

**Chignolin miniprotein** In this section, we report the results of our method obtained on a larger scale experiment: the folding/unfolding mechanism of the chignolin miniprotein. This system has

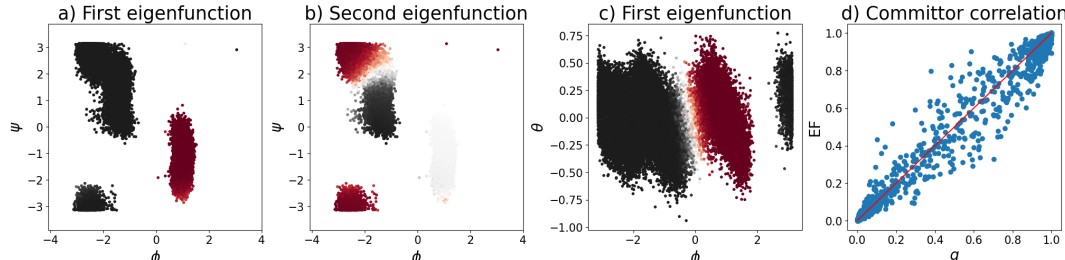

Figure 3: Alanine Dipeptide. Results of our method trained on Dataset 1 **a)** and **b)** first and second eigenfunctions represented on dataset 1, in the plane of the $\phi$ and $\psi$ dihedral angles. **c)** first eigenfunction represented on dataset 2, in the plane of the $\phi$ and $\theta$ dihedral angles, indicating that our method is effective even when trained from poor CVs (see text for more discussion). On all three panels, points are colored according to the value of the eigenfunction. **d)** Comparison of our method with the committor ($q$) of [19]

extensively been studied [46, 19, 39, 7]. We first performed a 1 $\mu$s biased simulation using the deep-TDA collective variable [46, 37] to gather transitions. Then we chose descriptors as input of the neural networks that are known to describe well the folding process [7]. Finally, we trained the method described in the current work with this trajectory and compared it with the results obtained when training on a $106\mu$s unbiased trajectory provided by D.E. Shaw research [28]. The results are presented in figure 4, showing a very good agreement between the training on an unbiased trajectory and on a biased one.

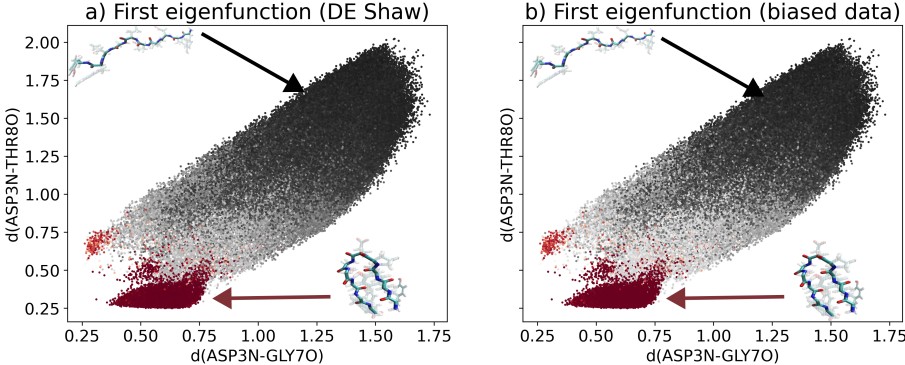

Figure 4: Our method for the chignolin miniprotein. The data points are represented in the plane of the distance between the nitrogen atom of the residue 3: ASP (ASP3N) and the oxygen atom of the residue 7: Gly (Gly7O) and the distance between ASP3N and the oxygen atom of residue 8: THR (THR8) which allow visualizing the folded and unfolded states.

## 7 Conclusions

We presented a method to learn the eigenfunctions and eigenvalues of the generator of Langevin dynamics from biased simulations, with strong theoretical guarantees. We contrasted this approach with those based on the transfer operator and a recent generator learning approach based on Rayleigh quotients. In experiments, we observed that our approach is effective even when trained from sub-optimal biased simulations. In the future our method could be applied to larger-scale simulations to discover rare events such as protein-ligand binding or catalytic processes. A main limitation of our method is that, in its current form, it is formulated for time-homogeneous bias potentials. However, the proposed framework could be naturally extended to time-dependent biasing, broadening its applicability in computational chemistry. Furthermore, given the quality of our results on alanine dipeptide, in the future, we can use our method to compute accurate eigenfunctions from old, possibly poorly converged, metadynamics simulations, thereby gaining novel and more accurate physical information.

## Acknowledgements

This work was partially funded by the European Union - NextGenerationEU initiative and the Italian National Recovery and Resilience Plan (PNRR) from the Ministry of University and Research (MUR), under Project PE0000013 CUP J53C22003010006 "Future Artificial Intelligence Research (FAIR)". We thank D.E Shaw research for providing the chignolin trajectory

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

# Appendix

The appendix contains additional background on stochastic differential equations, proofs of the results omitted in the main body, and more information about our learning method and the numerical experiments.

| notation | meaning | notation | meaning |
|---|---|---|---|
| $[\cdot]$ | set $\{1, 2 \ldots, \cdot\}$ | $[\cdot]_+$ | nonnegative part of a number |
| $\mathcal{X}$ | state space of the Markov process | $(X_t)_{t\geq 0}$ | time-homogeneous Markov process |
| $dW_t$ | Brownian motion | $\beta$ | inverse temperature of the system |
| $U$ | potential energy of the system | $V$ | bias potential |
| $\pi$ | Boltzmann distribution of potential $U$ | $\pi'$ | Boltzmann distribution of potential $U+V$ |
| $\widehat{\pi}$ | empirical version of $\pi$ | $\widehat{\pi}'$ | empirical version of $\pi'$ |
| $d\pi/d\pi'$ | density of $\pi$ w.r.t. $\pi'$ | $w$ | exponential weights of $d\pi/d\pi'$ |
| $L_\pi^2(\mathcal{X})$ | $L^2$ space on $\mathcal{X}$ w.r.t. the measure $\pi$ | $H_\pi^{1,2}(\mathcal{X})$ | Sobolev space w.r.t. $\pi$ on $\mathcal{X}$ |
| $\mathcal{T}_t$ | transfer operator with lag time $t$ | $\hat{T}_t$ | empirical estimator of $\mathcal{T}_t$ |
| $\mathcal{L}$ | generator of the true process | $\mathcal{L}'$ | generator of the biased process |
| $\eta$ | shift parameter | $(\eta I - \mathcal{L})^{-1}$ | resolvent at $\eta$ |
| $\mathcal{H}$ | hypothetical Hilbert space | $z$ | basis functions |
| $\mathfrak{E}^\eta[\cdot,\cdot]$ | regularized energy kernel | $\mathfrak{E}^\eta[\cdot]$ | regularized energy norm |
| $\mathcal{H}_\pi^\eta(\mathcal{X})$ | space associated to the energy norm | $\mathcal{Z}$ | injection operator |
| $\mathcal{R}_\partial$ | risk functional | $\widehat{\mathcal{R}}_\partial$ | empirical risk functional |
| C | covariance | $\widehat{\text{C}}$ | empirical covariance |
| W | covariance in energy space | $\widehat{\text{W}}$ | empirical covariance in energy space |
| $\lambda_i$ | generator eigenvalue | $f_i$ | generator eigenfunction |
| $\widehat{\lambda}_i$ | empirical estimator of $\lambda_i$ | $\widehat{f}_i$ | empirical estimator of $f_i$ |
| $z^\theta$ | neural network embedding | $\theta$ | neural network weights |
| $\mathcal{Z}_\theta$ | neural network injection operator | $\Lambda_\theta^\eta$ | neural network weights |
| $\mathcal{E}_\alpha$ | regularized loss function | $\widehat{\mathcal{E}}_\alpha$ | regularized empirical loss function |

Table 1: Summary of used notations.

## A   Background

In this work we consider stochastic differential equations (SDE) of the form

$$dX_t = a(X_t)dt + b(X_t)dW_t \quad \text{and} \quad X_0 = x. \tag{21}$$

The special case of Langevin equation considered in the main body of the paper corresponds to $a(x) = -\nabla U(x)$ and $b(x) = \sqrt{\frac{2}{\beta}}I$. Equation (21) describes the dynamics of the random vector $X_t$ in the state space $\mathcal{X} \subseteq \mathbb{R}^d$, governed by the *drift* $a : \mathbb{R}^d \to \mathbb{R}^d$ and the *diffusion* $b : \mathbb{R}^d \to \mathbb{R}^{d\times p}$ coefficients, where $W_t$ is a $\mathbb{R}^p$-dimensional standard Brownian motion. Under the usual conditions [see e.g. 33] that $a$ and $b$ are globally Lispchitz and sub-linear, the SDE (21) admits an unique strong solution $X = (X_t)_{\geq 0}$ that is a Markov process to which we can associate the semigroup of Markov *transfer operators* $(\mathcal{T}_t)_{t\geq 0}$ defined, for every $t \geq 0$, as

$$[\mathcal{T}_t f](x) := \mathbb{E}[f(X_t)|X_0 = x], \quad x \in \mathcal{X}, f \colon \mathcal{X} \to \mathbb{R}. \tag{22}$$

For stable processes, the distribution of $X_t$ converges to an *invariant measure* $\pi$ on $\mathcal{X}$, such that $X_0 \sim \pi$ implies that $X_t \sim \pi$ for all $t \geq 0$. In such cases, one can define the semigroup on $L_\pi^2(\mathcal{X})$, and characterize the process by the *infinitesimal generator* of the semi-group $(\mathcal{T}_t)_{t\geqslant 0}$,

$$\mathcal{L} := \lim_{t\to 0^+} \frac{\mathcal{T}_t - I}{t} \tag{23}$$

defined on the Sobolev space $H^{1,2}_\pi(\mathcal{X})$ of functions in $L^2_\pi(\mathcal{X})$ whose gradient are in $L^2_\pi(\mathcal{X})$, too, i.e. $\mathcal{L} \colon L^2_\pi(\mathcal{X}) \to L^2_\pi(\mathcal{X})$ and $\mathrm{dom}(\mathcal{L}) = H^{1,2}_\pi(\mathcal{X})$. The transfer operator and the generator are linked to each other by the formula $\mathcal{T}_t = \exp(t\mathcal{L})$.

After defining the infinitesimal generator for Markov processes by (23), we provide its explicit form for solution processes of equations like (21). Given a smooth function $f \in \mathcal{C}^2(\mathcal{X}, \mathbb{R})$, Itô's formula [see for instance 3, p. 495] provides for $t \in \mathbb{R}_+$,

$$f(X_t) - f(X_0) = \int_0^t \sum_{i=1}^d \partial_i f(X_s) dX_s^i + \tfrac{1}{2} \int_0^t \sum_{i,j=1}^d \partial^2_{ij} f(X_s) d\langle X^i, X^j \rangle_s$$

$$= \int_0^t \nabla f(X_s)^\intercal dX_s + \tfrac{1}{2} \int_0^t \mathrm{Tr}\big[X_s^\intercal (\nabla^2 f)(X_s) X_s\big] ds.$$

Recalling (21), we get

$$f(X_t) = f(X_0) + \int_0^t \left[ \nabla f(X_s)^\intercal a(X_s) + \tfrac{1}{2}\mathrm{Tr}\big[b(X_s)^\intercal (\nabla^2 f(X_s)) b(X_s)\big] \right] ds$$

$$+ \int_0^t \nabla f(X_s)^\intercal b(X_s) dW_s. \tag{24}$$

Provided $f$ and $b$ are smooth enough, the expectation of the last stochastic integral vanishes so that we get

$$\mathbb{E}[f(X_t)|X_0 = x] = f(x) + \int_0^t \mathbb{E}\Big[\nabla f(X_s)^\intercal a(X_s) + \tfrac{1}{2}\mathrm{Tr}\big[b(X_s)^\intercal (\nabla^2 f(X_s)) b(X_s)\big]\Big|X_0 = x\Big] ds$$

Recalling that $\mathcal{L} = \lim_{t \to 0^+} (\mathcal{T}_t f - f)/t$, we get for every $x \in \mathcal{X}$,

$$\mathcal{L}f(x) = \lim_{t \to 0} \frac{\mathbb{E}[f(X_t)|X_0 = x] - f(x)}{t}$$

$$= \lim_{t \to 0} \frac{1}{t}\left[ \int_0^t \mathbb{E}\Big[\nabla f(X_s)^\intercal a(X_s) + \tfrac{1}{2}\mathrm{Tr}\big[(X_s)^\intercal (\nabla^2 f(X_s)) b(X_s)\big]\Big] ds \Big| X_0 = x \right]$$

$$= \nabla f(x)^\intercal a(x) + \tfrac{1}{2}\mathrm{Tr}\big[b(x)^\intercal (\nabla^2 f(x)) b(x)\big], \tag{25}$$

which provides the closed formula for the IG associated with the solution process of (21). In particular, for Langevin dynamics this reduces to (3).

Next, recalling that for a bounded linear operator $A$ on some Hilbert space $\mathcal{H}$ the *resolvent set* of the operator $A$ is defined as $\rho(A) = \{\lambda \in \mathbb{C} \,|\, A - \lambda I \text{ is bijective}\}$, and its *spectrum* $\mathrm{Sp}(A) = \mathbb{C}\backslash\{\rho(A)\}$, let $\lambda \subseteq \mathrm{Sp}(A)$ be the isolated part of the spectra, i.e. both $\lambda$ and $\mu = \mathrm{Sp}(A) \setminus \lambda$ are closed in $\mathrm{Sp}(A)$. Then, the *Riesz spectral projector* $P_\lambda \colon \mathcal{H} \to \mathcal{H}$ is defined by

$$P_\lambda = \frac{1}{2\pi} \int_\Gamma (zI - A)^{-1} dz, \tag{26}$$

where $\Gamma$ is any contour in the resolvent set $\mathrm{Res}(A)$ with $\lambda$ in its interior and separating $\lambda$ from $\mu$. Indeed, we have that $P_\lambda^2 = P_\lambda$ and $\mathcal{H} = \mathrm{Im}(P_\lambda) \oplus \mathrm{Ker}(P_\lambda)$ where $\mathrm{Im}(P_\lambda)$ and $\mathrm{Ker}(P_\lambda)$ are both invariant under $A$, and we have $\mathrm{Sp}(A_{|\mathrm{Im}(P_\lambda)}) = \lambda$ and $\mathrm{Sp}(A_{|\mathrm{Ker}(P_\lambda)}) = \mu$. Moreover, $P_\lambda + P_\mu = I$ and $P_\lambda P_\mu = P_\mu P_\lambda = 0$.

Finally if $A$ is a *compact* operator, then the Riesz-Schauder theorem [see e.g. 38] assures that $\mathrm{Sp}(T)$ is a discrete set having no limit points except possibly $\lambda = 0$. Moreover, for any nonzero $\lambda \in \mathrm{Sp}(T)$, then $\lambda$ is an *eigenvalue* (i.e. it belongs to the point spectrum) of finite multiplicity, and, hence, we can deduce the spectral decomposition in the form

$$A = \sum_{\lambda \in \mathrm{Sp}(A)} \lambda P_\lambda, \tag{27}$$

where the geometric multiplicity of $\lambda$, $r_\lambda = \mathrm{rank}(P_\lambda)$, is bounded by the algebraic multiplicity of $\lambda$. If additionally $A$ is a normal operator, i.e. $AA^* = A^*A$, then $P_\lambda = P_\lambda^*$ is an orthogonal projector for each $\lambda \in \mathrm{Sp}(A)$ and $P_\lambda = \sum_{i=1}^{r_\lambda} \psi_i \otimes \psi_i$, where $\psi_i$ are normalized eigenfunctions of $A$ corresponding to $\lambda$ and $r_\lambda$ is both algebraic and geometric multiplicity of $\lambda$.

We conclude this section by stating the well-known Davis-Kahan perturbation bound for eigenfunctions of self-adjoint compact operators.

**Proposition 1** ([13]). *Let $A$ be compact self-adjoint operator on a separable Hilbert space $\mathcal{H}$. Given a pair $(\widehat{\mu}, \widehat{f}) \in \mathbb{C} \times \mathcal{H}$ such that $\|\widehat{f}\| = 1$, let $\lambda$ be the eigenvalue of $A$ that is closest to $\widehat{\mu}$ and let $f$ be its normalized eigenfunction. If $\widehat{g} = \min\{|\widehat{\mu} - \lambda| \mid \lambda \in \mathrm{Sp}(A) \setminus \{\lambda\}\} > 0$, then $\sin(\sphericalangle(\widehat{f}, f)) \leq \|A\widehat{f} - \widehat{\mu}\widehat{f}\|/\widehat{g}$.*

# B Unbiased generator regression

In this section, we prove Theorem 1 relying on recently developed statistical theory of generator learning [21]. To that end, let $\phi(x) := z(\cdot)^{\mathsf{T}} z(x) \in \mathcal{H}$ be a feature map of the RKHS space $\mathcal{H}$ of dimension $\dim(\mathcal{H}) = m$. Let $\mathrm{W}u_j = \sigma_j^2 u_j$ be the eigenvalue decomposition of $\mathrm{W}$ and let $v_j := u_j/\sigma_j$. This induces the SVD of the injection operator, $\mathcal{Z}u_j = \sigma_j \tilde{z}_j$ for $\tilde{z}_j := z(\cdot)^{\mathsf{T}} v_j$.

Since $\mathcal{H} \subseteq H_\pi^{1,\infty}(\mathcal{X})$, we have that

$$c_\tau = \operatorname*{ess\,sup}_{x \sim \pi} \sum_{j \in \mathbb{N}} [\eta|\tilde{z}_j(x)|^2 - z_j(x)[\mathcal{L}\tilde{z}_j](x)] < +\infty,$$

and, denoting $\mathrm{W}_\gamma := \mathrm{W} + \eta\gamma\mathrm{I}$

$$\mathrm{tr}[\mathrm{W}_\gamma^{-1}\mathrm{W}_\gamma] \leq m.$$

In addition denote the empirical version of $\mathrm{W}_\gamma$ as $\widehat{\mathrm{W}}_\gamma := \widehat{\mathrm{W}} + \eta\gamma\mathrm{I}$.

Now, we can apply the following propositions from [21] to our setting, recalling the notation for normalizing constant $\overline{w} := \mathbb{E}_{x \sim \pi'}[w(x)]$ for which $w(\cdot)/\overline{w} = d\pi/d\pi'$.

**Proposition 2.** *Given $\delta > 0$, with probability in the i.i.d. draw of $(x_i)_{i=1}^n$ from $\pi$, it holds that*

$$\mathbb{P}\{\|\widehat{\mathrm{W}} - \mathrm{W}\| \leq \varepsilon_n(\delta)\} \geq 1 - \delta,$$

*where*

$$\varepsilon_n(\delta) = \frac{2\|\mathrm{W}\|}{3n}\mathcal{L}(\delta) + \sqrt{\frac{2\|\mathrm{W}\|}{n}\mathcal{L}(\delta)} \quad \text{and} \quad \mathcal{L}(\delta) = \log\frac{4\,\mathrm{tr}(\mathrm{W})}{\delta\,\|\mathrm{W}\|}. \tag{28}$$

**Proposition 3.** *Given $\delta > 0$, with probability in the i.i.d. draw of $(x_i)_{i=1}^n$ from $\pi$, it holds that*

$$\mathbb{P}\left\{\|\mathrm{W}_\gamma^{-1/2}(\widehat{\mathrm{W}} - \mathrm{W})\mathrm{W}_\gamma^{-1/2}\| \leq \varepsilon_n^1(\gamma, \delta)\right\} \geq 1 - \delta, \tag{29}$$

*where*

$$\varepsilon_n^1(\gamma, \delta) = \frac{2c_\tau}{3n}\mathcal{L}^1(\gamma, \delta) + \sqrt{\frac{2\,c_\tau}{n}\mathcal{L}^1(\gamma, \delta)}, \tag{30}$$

*and*

$$\mathcal{L}^1(\gamma, \delta) = \ln\frac{4}{\delta} + \ln\frac{\mathrm{tr}(\mathrm{W}_\gamma^{-1}\mathrm{W})}{\|\mathrm{W}_\gamma^{-1}\mathrm{W}\|}.$$

*Moreover,*

$$\mathbb{P}\left\{\|\mathrm{W}_\gamma^{1/2}\widehat{\mathrm{W}}_\gamma^{-1}\mathrm{W}_\gamma^{1/2}\| \leq \frac{1}{1 - \varepsilon_n^1(\gamma, \delta)}\right\} \geq 1 - \delta. \tag{31}$$

**Proposition 4.** *With probability in the i.i.d. draw of $(x_i)_{i=1}^n$ from $\pi$, it holds*

$$\mathbb{P}\left\{\|\mathrm{W}_\gamma^{-1/2}(\widehat{\mathrm{C}} - \mathrm{C})\|_F \leq \varepsilon_n^2(\gamma, \delta)\right\} \geq 1 - \delta,$$

*where*

$$\varepsilon_n^2(\gamma, \delta) = \frac{4\sqrt{2\,m\|\mathrm{W}\|}}{\eta}\ln\frac{2}{\delta}\sqrt{\frac{c_\beta}{n} + \frac{c_\tau}{n^2}}. \tag{32}$$

We are now ready to prove Theorem 1, which we restate here for convenience.

**Theorem 1.** *Let $\mathcal{D}_n = (x_i')_{i \in [n]}$ be the biased dataset generated from $\pi'$. Let $w(x) = e^{\beta V(x)}$ and define the empirical covariances w.r.t. the empirical distribution $\widehat{\pi}' = n^{-1}\sum_{i \in [n]} \delta_{x_i'}$ by*

$$\widehat{\mathrm{C}} = \left(\mathbb{E}_{x' \sim \widehat{\pi}'}[w(x')z_i(x')z_j(x')]\right)_{i,j \in [m]} \quad \text{and} \quad \widehat{\mathrm{W}} = \left(\mathfrak{E}_{\widehat{\pi}'}^\eta[\sqrt{w}z_i, \sqrt{w}z_j]\right)_{i,j \in [m]}. \tag{17}$$

*Compute the eigenpairs $(\nu_i, v_i)_{i\in[m]}$ of the RR estimator $\widehat{G}_{\eta,\gamma} = (\widehat{W} + \eta\gamma I)^{-1}\widehat{C}$, and estimate the eigenpairs in (4) as $(\widehat{\lambda}_i, \widehat{f}_i) = (\eta - 1/\nu_i, z(\cdot)^\top v_i)$. If the elements of $\mathcal{H}$ and their gradients are essentially bounded, and $\lim_{m\to\infty} \rho(\mathcal{H}) = 0$, then for every $\varepsilon > 0$, there exist $(m, n, \gamma) \in \mathbb{N} \times \mathbb{N} \times \mathbb{R}_+$, such that, for every $i \in [m]$, $|\lambda_i - \widehat{\lambda}_i| \le \varepsilon$ and $\sin_{L^2_\pi}(\triangleleft(f_i, \widehat{f}_i)) \le \varepsilon$, with high probability.*

*Proof.* We first show that $\mathcal{R}_\partial(\widehat{G}_{\eta,\gamma}) < \varepsilon$ for big enough $m, n \in \mathbb{N}$ and small enough $\gamma > 0$.

Observe that

$$\mathbf{W} = \mathbb{E}[\widehat{\mathbf{W}}]/\overline{w} \quad \text{and} \quad \mathbf{C} = \mathbb{E}[\widehat{\mathbf{C}}]/\overline{w} \tag{33}$$

and, hence

$$\mathbf{G}_{\eta,\gamma} := \mathbf{W}_\gamma^{-1}\mathbf{C} = (\mathbb{E}[\widehat{\mathbf{W}}_\gamma])^{-1}(\mathbb{E}[\widehat{\mathbf{C}}]),$$

due to cancellation of $\overline{w}$.

Given $\varepsilon > 0$, let $m \in \mathbb{N}$ be such that $\rho(\mathcal{H}) = \|(I - P_\mathcal{H})(\eta I - \mathcal{L})^{-1}\|^2_{\mathrm{HS}(\mathcal{H}, \mathcal{H}^\eta_\pi)} < \varepsilon/3$. Next, since

$$P_\mathcal{H}(\eta I - \mathcal{L})^{-1}\mathcal{Z} - \mathcal{Z}\mathbf{G}_{\eta,\gamma} = (I - \mathcal{Z}\mathbf{W}_\gamma^{-1}\mathcal{Z}^*)(\eta I - \mathcal{L})^{-1}\mathcal{Z} = \mathcal{Z}(\mathbf{W}^\dagger\mathbf{C} - \mathbf{W}_\gamma^{-1}\mathbf{C}),$$

we have that

$$\|P_\mathcal{H}(\eta I - \mathcal{L})^{-1}\mathcal{Z} - \mathcal{Z}\mathbf{G}_{\eta,\gamma}\|_{\mathrm{HS}(\mathcal{H}, \mathcal{H}^\eta_\pi)} = \|\mathbf{W}^{1/2}(\mathbf{W}^\dagger\mathbf{C} - \mathbf{W}_\gamma^{-1}\mathbf{C})\|_F = \|\mathbf{W}^{1/2}(Wx^\dagger - \mathbf{W}_\gamma^{-1})\mathbf{C}\|_F \to 0,$$

as $\gamma \to 0$. Hence, let $\gamma > 0$ be such that $\|P_\mathcal{H}(\eta I - \mathcal{L})^{-1}\mathcal{Z} - \mathcal{Z}\mathbf{G}_{\eta,\gamma}\|_{\mathcal{H}\to\mathcal{H}^\eta_\pi} < \varepsilon/3$.

Finally, using the decomposition of the risk

$$\mathcal{R}_\partial(\widehat{G}_{\eta,\gamma}) \le \rho(\mathcal{H}) + \|P_\mathcal{H}(\eta I - \mathcal{L})^{-1}\mathcal{Z} - \mathcal{Z}\mathbf{G}_{\eta,\gamma}\|_{\mathrm{HS}(\mathbb{R}^m, \mathcal{H}^\eta_\pi)} + \|\mathcal{Z}(\widehat{G}_{\eta,\gamma} - \widehat{G}_{\eta,\gamma})\|_{\mathrm{HS}(\mathbb{R}^m, \mathcal{H}^\eta_\pi)}$$

it remains to show that for large enough $n$ we have $\|\mathcal{Z}(\mathbf{G}_{\eta,\gamma} - \widehat{G}_{\eta,\gamma})\|_{\mathrm{HS}(\mathbb{R}^m, \mathcal{H}^\eta_\pi)} \le \varepsilon/3$.

To that end observe that

$$\begin{aligned}
\mathbf{W}_\gamma^{1/2}(\widehat{G}_{\eta,\gamma} - \mathbf{G}_{\eta,\gamma}) &= \mathbf{W}_\gamma^{1/2}\widehat{\mathbf{W}}_\gamma^{-1}(\widehat{\mathbf{C}} - \widehat{\mathbf{W}}_\gamma\mathbf{W}_\gamma^{-1}\mathbf{C} \pm \mathbf{C}) \\
&= \mathbf{W}_\gamma^{1/2}\widehat{\mathbf{W}}_\gamma^{-1}\mathbf{W}_\gamma^{1/2}\left(\mathbf{W}_\gamma^{-1/2}(\widehat{\mathbf{C}} - \mathbf{C}) - \mathbf{W}_\gamma^{-1/2}(\widehat{\mathbf{W}} - \mathbf{W})\mathbf{W}_\gamma^{-1/2}(\mathbf{W}_\gamma^{-1/2}\mathbf{C})\right).
\end{aligned}$$

Thus, by multiplying the above expression by $\overline{w}$ and applying Propositions 3 and 4, we obtain that there exists $n \in \mathbb{N}$ such that $\|\mathcal{Z}(\mathbf{G}_{\eta,\gamma} - \widehat{G}_{\eta,\gamma})\|_{\mathrm{HS}(\mathbb{R}^m, \mathcal{H}^\eta_\pi)} \le \varepsilon/3$.

Next, assuming that $\|\widehat{\mathbf{W}} - \mathbf{W}\|$ is small, for the normalization of the estimated eigenfunctions we have that

$$\frac{\|v_j\|^2_2}{\|\widehat{f}_j\|^2_{\mathcal{H}^\eta_\pi}} = \frac{v_j^\top v_j}{v_j^\top \mathbf{W} v_j} \le \frac{v_j^\top v_j}{v_j^\top\widehat{\mathbf{W}}v_j - v_j^\top(\mathbf{W} - \widehat{\mathbf{W}})v_j} \le \frac{1}{\lambda^+_{\min}(\widehat{\mathbf{W}}) - \|\widehat{\mathbf{W}} - \mathbf{W}\|} \le \frac{1}{\lambda_m(\mathbf{W}) - 2\|\widehat{\mathbf{W}} - \mathbf{W}\|}.$$

where we have that $\lambda_m(\mathbf{W}) > 0$ due to fact that $(z_j)$ are linearly independent.

Therefore, to conclude the proof, we apply [21, Proposition 2] which directly relying on Proposition 1 yields the result. $\qquad\square$

At last we remark, based on the observation that $\mathbf{W} = \mathbb{E}[\widehat{\mathbf{W}}]/\overline{w}$ and $\mathbf{C} = \mathbb{E}[\widehat{\mathbf{C}}]/\overline{w}$, one can readily obtain stronger version of Theorem 1 in the general RKHS setting of [21].

# C  Unbiased deep learning of spectral features

In this section, we provide details on our DNN method and prove Theorem 2. To that end, let us denote the terms in the loss as

$$\mathcal{E}_\gamma(\theta) := \underbrace{\|(\eta I - \mathcal{L})^{-1} - \mathcal{Z}_\theta\Lambda^\theta_\eta\mathcal{Z}^*_\theta\|^2_{\mathrm{HS}(\mathcal{H}^\eta_\pi)} - \|(\eta I - \mathcal{L})^{-1}\|^2_{\mathrm{HS}(\mathcal{H}^\eta_\pi)}}_{\text{expected loss } \mathcal{E}} + \gamma\underbrace{\sum_{i,j\in[m]}(\langle z^\theta_i, z^\theta_j\rangle_{L^2_\pi} - \delta_{i,j})^2}_{\text{orthonormality loss } \mathcal{E}_{\mathrm{on}}}, \tag{34}$$

and recall that the injection operator is $\mathcal{Z}_\theta = (z^\theta)^\intercal(\cdot)\colon \mathbb{R}^m \to \mathcal{H}_\pi^\eta(\mathcal{X})$. It is easy to show from the definition of the adjoint that, for every $f \in \mathcal{H}_\pi^\eta(\mathcal{X})$, we have

$$\mathcal{Z}_\theta^* f = \mathbb{E}_{x \sim \pi}[z^\theta(x)((\eta I - \mathcal{L})f)(x)]. \tag{35}$$

Thus, $\mathcal{Z}_\theta^* \mathcal{Z}_\theta = \mathfrak{C}^\eta[z_i^\theta, z_j^\theta]_{i,j \in [m]}$ which we denote by $\mathrm{W}_\theta$, while $\mathcal{Z}_\theta^*(\eta I - \mathcal{L})\mathcal{Z}_\theta = \mathbb{E}_{x \sim \pi}[z_i^\theta(x) z_j^\theta(x)]_{i,j \in [m]}$ is denoted by $\mathrm{C}_\theta$.

**Theorem 2.** *Given a compact operator $(\eta I - \mathcal{L})^{-1}$, $\eta > 0$, if $(z^\theta)_{i \in [m]} \subseteq \mathcal{H}_\pi^\eta(\mathcal{X})$ for all $\theta \in \Theta$, then*

$$\mathbb{E}\left[\mathcal{E}_\alpha^{\widehat{\pi}_1', \widehat{\pi}_2'}(\theta)\right] = \overline{w}^2 \, \mathcal{E}_\alpha(\theta) \geq -\sum_{i \in [m]} \tfrac{\overline{w}^2}{(\eta - \lambda_i)^2}, \quad \textit{for all } \theta \in \Theta, \tag{20}$$

*where $\overline{w} = \mathbb{E}_{x \sim \pi'}[w(x)]$. Moreover, if $\alpha > 0$ and $\lambda_{m+2} < \lambda_{m+1}$, then the equality holds if and only if $(\lambda_i^\theta, z_i^\theta) = (\lambda_i, f_i)$ $\pi$-a.e., up to the ordering of indices and choice of eigenfunction signs for $i \in [m]$.*

*Proof.* Let $P_k\colon \mathcal{H}_\pi^\eta(\mathcal{X}) \to \mathcal{H}_\pi^\eta(\mathcal{X})$ be spectral projector of $\mathcal{L}$ corresponding to the $k$ largest eigenvalues. Now, consider

$$\mathcal{E}^k(\theta) = \|P_k(\eta I - \mathcal{L})^{-1} - \mathcal{Z}_\theta \Lambda_\eta^\theta \mathcal{Z}_\theta^*\|_{\mathrm{HS}(\mathcal{H}_\pi^\eta)}^2 - \|P_k(\eta I - \mathcal{L})^{-1}\|_{\mathrm{HS}(\mathcal{H}_\pi^\eta)}^2.$$

Due to Eckhart-Young theorem, we have that for every $\theta \in \Theta$, the best rank-$m$ approximation of $(\eta I - \mathcal{L})^{-1}$ is $(\eta I - \mathcal{L})^{-1} P_m = (\eta I - \mathcal{L})^{-1} P_k P_m$, for $k > m$, and it holds

$$\mathcal{E}^k(\theta) \geq \sum_{j=m+1}^{k}(\eta - \lambda_i)^{-1} - \sum_{j=1}^{k}(\eta - \lambda_i)^{-1} = -\sum_{j=1}^{m}(\eta - \lambda_i)^{-1}.$$

As before, expanding in $\mathcal{E}^k$ the HS norm via the trace, we obtain

$$\mathcal{E}^k(\theta) = \|\mathcal{Z}_\theta \Lambda_\eta^\theta \mathcal{Z}_\theta^*\|_{\mathrm{HS}(\mathcal{H}_\pi^\eta)}^2 - 2\operatorname{tr}[\mathcal{Z}_\theta \Lambda_\eta^\theta \mathcal{Z}_\theta^*(\eta I - \mathcal{L})^{-1} P_k] = \mathcal{E}(\theta) + 2\operatorname{tr}[\mathcal{Z}_\theta \Lambda_\eta^\theta \mathcal{Z}_\theta^*(\eta I - \mathcal{L})^{-1}(I - P_k)],$$

and, hence, by Cauchy-Schwartz inequality, we have that

$$|\mathcal{E}(\theta) - \mathcal{E}^k(\theta)| \leq \|\mathcal{Z}_\theta\|_{\mathrm{HS}(\mathcal{H}, \mathcal{H}_\pi^\eta)}^2 \|\Lambda_\eta^\theta\| \|(\eta I - \mathcal{L})^{-1}(I - P_k)\| = \tfrac{1}{\eta} \sum_{i \in [m]} \mathfrak{C}_\pi^\eta[z_i^\theta](\eta - \lambda_{k+1})^{-1}.$$

Observing that $z_i^\theta \in \mathcal{H}_\pi^\eta(\mathcal{X})$, i.e. $\mathfrak{C}_\pi^\eta[z_i^\theta] < \infty$, we conclude that, for every $\theta \in \Theta$, $\lim_{k \to \infty} \mathcal{E}^k(\theta) = \mathcal{E}(\theta)$. Therefore, noting that $\mathcal{E}_\alpha(\theta) \geq \mathcal{E}(\theta)$, inequality in (20) is proven. Since the equality clearly holds for the leading eigenpairs of the generator, to prove the reverse, it suffices to recall the uniqueness result of the best rank-$m$ estimator, which is given by $P_m(\eta I - \mathcal{L})^{-1}$, i.e. $(z_i^\theta)_{i \in [m]}$ span the leading invariant subspace $(f_i)_{i \in [m]}$ of the generator. So, if

$$\mathcal{E}_\alpha(\theta) = \mathcal{E}(\theta) = \sum_{j=1}^{m}(\eta - \lambda_i)^{-1}$$

and $\alpha > 0$, we have that $\mathcal{E}_{\mathrm{on}}(\theta) = 0$, implying that $(z_i)_{i \in [m]}$ is an orthonormal basis, and, hence $P_m(\eta I - \mathcal{L})^{-1}$, i.e. $(z_i^\theta)_{i \in [m]} = \mathcal{Z}_\theta \Lambda_\theta^\eta \mathcal{Z}_\theta^*$. The result follows.

To show that

$$\mathcal{E}_\gamma(\theta) = \mathbb{E}\left[\mathcal{E}_\gamma^{\widehat{\pi}_1', \widehat{\pi}_2'}(\theta)\right] / \overline{w}^2,$$

we rewrite (18) to encounter the distribution change, noting that the empirical covariances are reweighted but not normaliezed by $\overline{w}$. So, we have that

$$\mathcal{E}_\gamma(\theta) = \operatorname{tr}\left[\overline{w}^{-2} \mathbb{E}[\widehat{\mathrm{C}}_\theta] \Lambda_\theta^\eta \mathbb{E}[\widehat{\mathrm{W}}_\theta] \Lambda_\theta^\eta - 2\overline{w}^{-1} \mathbb{E}[\mathrm{C}_\theta] \Lambda_\theta^\eta + \overline{w}^{-2} \alpha(\mathbb{E}[\widehat{\mathrm{C}}_\theta] - \overline{w}\mathrm{I})^2\right].$$

But, since

$$\mathbb{E}_{x \sim \pi}[f(x)g(x)] = \mathbb{E}_{x' \sim \pi'}\left[\tfrac{d\pi}{d\pi'}(x') \, f(x') \, g(x')\right] = \mathbb{E}_{x_1' \sim \pi'}\left[\sqrt{\tfrac{d\pi}{d\pi'}(x_1')} f(x_1')\right] \mathbb{E}_{x_2' \sim \pi'}\left[\sqrt{\tfrac{d\pi}{d\pi'}(x_2')} g(x_2')\right],$$

where $x_1'$ and $x_2'$ are two independent r.v. with a law $\pi'$, the proof is completed. $\qquad\square$

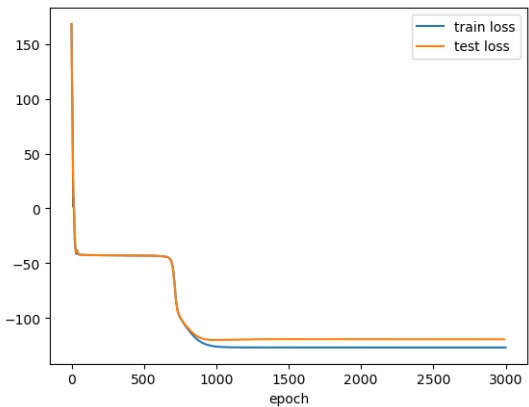

Figure 5: Typical behavior of the loss function during a training.

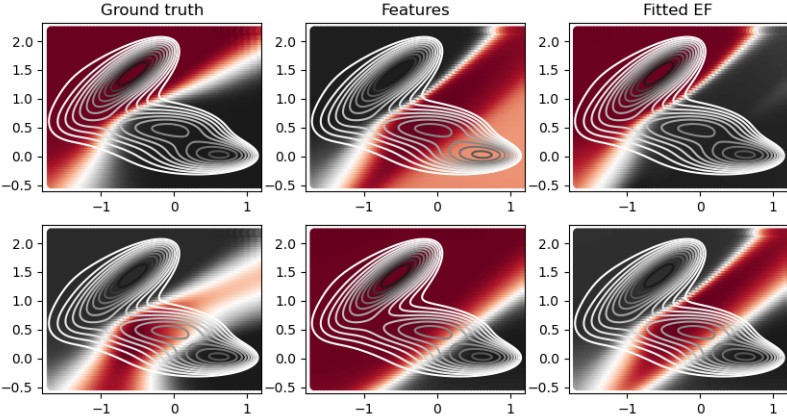

Figure 6: Comparison on Muller Brown potential with ground truth, learned features and fitted eigenfunctions

# D   Training of a neural network model

## D.1   Evolution of the loss with eigenfunctions

It is interesting to note that during the training process, the loss reaches progressively lower plateaus. This is due to the fact that the NN has found a novel eigenfunction orthogonal to the previously found ones, starting with the constant one. Then during the plateau phase, the subspace is being explored until a new relevant direction is found. Typical behavior is shown in Figure 5. It was obtained for the case of a double well potential (see appendix below), but the same behavior was observed in all the training sessions. This nice property is a handy tool in properly optimizing the loss and understanding the proper stopping time.

## D.2   Training with imperfect features

One of the main advantages of our method is that even with features that are not eigenfunctions of the generator, but that were trained with our method, we can recover the good eigenfunction estimates as proven in Theorem 1. In Figure 6, we illustrate such situation on a simulation of the Muller Brown potential: the trained features do not represent the ground truth, however, using our fitting method on the same dataset, we managed to recover eigenfunctions close to the ground truth. This is to the best of our knowledge the first time this kind of "learn and fit" method has been applied to the learning of the infinitesimal generator.

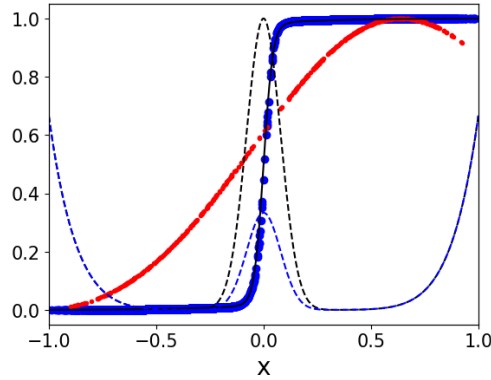

Figure 7: Simulation of the double well potential (black dashed lines) under an effective biased potential (blue dashed lines). Our method (blue points) compared to ground truth (black line) and transfer operator based approach (red points)

### D.3 Activation functions and structure of the neural network

In all of our experiments we used the hyperbolic tangent activation function. This choice was made because it is a widely used, bounded function with continuous derivative. It thus satisfies all the criteria needed for this method. Finally, when looking for $m$ eigenpairs, instead of having a neural network with $m$ outputs, we choose to have $m$ neural networks with one output.

### D.4 Hyperparameters

Besides common hyperparameters such as learning rate, neural network architecture and activation function, our method requires only two hyperparameters: $\eta$ and $\alpha$. Other methods such as [50] do not require $\eta$, but on the other hand requires one weight per searched eigenfunction.

## E Experiments

For all the experiments we used pytorch 1.13, and the optimizations of the models were performed using the ADAM optimizer. The version of python used is 3.9.18. All the experiments were performed on a workstation with a AMD® Ryzen threadripper pro 3975wx 32-cores processor and an NVIDIA Quadro RTX 4000 GPU. In all the experiments, the datasets were randomly split into a training and a validation dataset. The proportion were set to 80% for training and 20% for validation. The training of deepTICA models was performed using the mlcolvar package [8].

### E.1 One dimensional double well potential

In this subsection, we showcase the efficiency of our method on a simple one dimensional toy model.

The target potential we want to sample has the form $U_{\text{tg}}(x) = 4(-1.5 \exp(-80x^2) + x^8)$, which has a form of two wells separated by a high barrier which can hardly be crossed during a simulation. In order to observe more transitions between the two wells and efficiently sample the space, we lower the barrier by running simulations under the following potential: $U_{\text{sim}} = 4(-0.5 \exp(-80x^2) + x^8)$, which thus makes a bias potential: $V_{\text{bias}}(x) = U_{\text{sim}}(x) - U_{\text{tg}}(x) = -4(\exp(-80x^2))$. In Figure 7, we compare our method based on kernel methods (infinite dimensional dictionary of functions) with the ground truth and transfer operator baselines, namely deepTICA [7] which is a state-of-the-art method for molecular dynamics simulations. For this experiment we have used a Gaussian kernel of lengthscale 0.1, $\eta = 0.1$ and a regularization parameter of $10^{-5}$

### E.2 Muller Brown potential

For this experiment, the dataset was generated using an in-house code implementing the Euler-Maruyama scheme to discretize the overdamped Langevin equation. The simulation was performed at

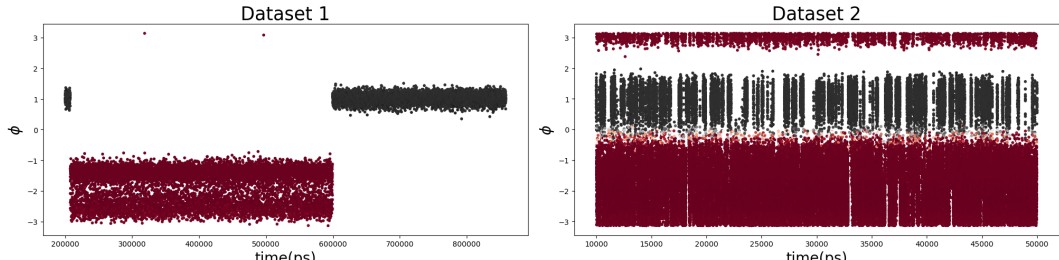

Figure 8: Time evolution of the $\phi$ angle in both datasets. Points are colored according to the value of the first nontrivial eigenfunction

a temperature of 1 (arbitrary unit) and a timestep of $10^{-3}$. The simulation was run for $10^7$ timesteps. The bias potential is built according to the following equation

$$U(x,t) = h \sum_{i=1}^{N_t} e^{-\frac{\|x-x_i\|^2}{2\sigma^2}} \tag{36}$$

where the centers $x_i$ are built on the fly: every 500 timestep one more center is added to this list. This kind of bias potential is called metadynamics [25] and allows reducing the height of the barrier for a better exploration of the space. In order to have a static potential, no more centers are added after 300000 timesteps. We use a learning rate of $5.10^{-3}$, the architecture of the neural network used is a multilayer perceptron with layers of size 2 (inputs), 20, 20 and 1. The parameter $\eta$ was chosen to be 0.05.

### E.3  Alanine dipeptide

**Simulation details**   All the simulations are run with GROMACS 2022.3 [2] and patched with plumed 2.10 [45] in order to perform enhanced sampling simulations. We used the Amber99-SB [40] force field. The Langevin dynamics was sampled with a timestep of 2fs with a damping coefficent $\gamma_i = m_i/0.05$ at a target temperture of 300K. For both dataset, in order to make proper comparison, we used the explore version of OPES [17], with a barrier parameter of 20 kJ/mol and a pace of deposition of 500 timesteps.

**Neural networks training**   For our neural networks training, we assume overdamped Langevin dynamics with the same value of the friction coefficient as in the simulation, as done in other works [50]. We use a learning rate of $10^{-3}$, and the architecture of the neural network used is a multilayer perceptron with layers of size 30 (inputs), 20, 20 and 1. The parameter $\eta$ was chosen to be 0.1.

### E.4  Chignolin

**Simulation details**   The simulations are run with GROMACS 2022.3 [2] and patched with plumed 2.10 [45]. They share the same setup used for the D.E Shaw trajectory [28] used in the main text . For the same reason, we kept the simulation condition consistent with that work. All simulations were performed with an integration time step of 2 fs and sampling NVT ensemble at 340K. The deepTDA model used for biasing the simulation is the one obtained in ref [46]

**Neural networks training**   For our neural networks training, we assume overdamped Langevin dynamics. We use a learning rate of $5.10^{-4}$, and the architecture of the neural network used is a multilayer perceptron with layers of size 210 (inputs), 50, 50 and 1. The parameter $\eta$ was chosen to be 0.2.

