# OpenReview forum: "From Biased to Unbiased Dynamics: An Infinitesimal Generator Approach"
_NeurIPS.cc/2024/Conference — NeurIPS 2024 poster_

### Official Review · Reviewer_BQ7h · 2024-07-10

**Soundness:** 4
**Presentation:** 2
**Contribution:** 4
**Rating:** 8
**Confidence:** 3

**Summary:**

@Authors, I would appreciate it if you could point out any inaccuracies in the following summary since it took me a long time to understand your paper and am still not completely certain my understanding is correct.

The paper aims to address a common problem for molecular dynamics simulations, which is to obtain collective variables that can be used to bias simulations such that, e.g., rare transitions occur more frequently and important observables can be computed faster. The authors follow previous work and learn the system's generator, whose eigenvalues inform transition timescales and whose eigenfunctions can be used as collective variables to bias simulations along. The novelty is in deriving an approach that allows learning the generator from biased enhanced sampling simulations which converge faster and observe more transitions than conventional simulation. The authors evaluate their method on toy systems.

**Strengths:**

1. Novel approach to accelerate obtaining scientifically important collective variables by enabling their computation from simulations biased by enhanced sampling. This approach requires developing new theory and to derive a training objective. The procedure to do so seems non trivial to me but I cannot sufficiently judge the value of the provided theory as I can only follow the broad strokes and am not well versed in the used math.
   I hope other reviewers can provide more useful signal for this aspect.
2. Impactful application: The paper provides useful ideas for a problem with significance for scientific applications that can have large downstream impacts.
3. Very well written introduction and overview of the transfer operator and generator formalism.

**Weaknesses:**

My points contain many understanding questions and some "criticizing" questions. It would be great if you could also answer the understanding questions.
1. Experiments: The experiments are carried out on three toy systems and the proposed approach visually improves upon two deep learning baselines in one of the three experiments.
	1. I understand that the cited works on generator learning provide the same or fewer experiments. However, from skimming the cited works it also seems to me that one of the main areas of application would be simulations on proteins. Is that incorrect and if not, why do you and others not provide experiments on proteins? Aren't there proteins with well known dynamics on which we could evaluate the methods?
	2. **I think this method should be tried on systems of increasing sizes until it fails. The experiments of a paper introducing a new method ideally provide information on the capabilities of a method.**  Why do you (and the rest of the generator learning papers) not do so?
	3. Why do you omit the deep learning baselines from the ALDP experiments?
2. For an understanding of how your method fits into the broader landscape of approaches to determine CVs, I think it would be important to also provide comparisons with non-deep learning approaches. The relationship to DL methods is clear but to understand the general "usefulness" it would be nice to have delineation from classical methods in terms of approach and experiments. Are there classical approaches that can be used as baselines that would outperform all DL methods?
3. Speed comparisons: it seems to me that the underlying tradeoff for all methods is between speed and accuracy and that accuracy alone might be less meaningful. Is there nothing to be said about the runtimes for data collection and training?
Minor:
1. To make the paper more broadly accessible to the wider deep learning community, I think it would be helpful to describe your final operator learning approach more procedurally on e.g. the ALDP example. What is the neural network input and output dimensionality? Why do you have separate neural networks for each operator output dimension? What is the input and output to your neural network?
   Once you trained your neural network, how do you obtain the eigenfunctions from it.
2. How do you obtain your plots in e.g. the ALDP figure Figure 2.? To understand this: what concretely is the eigenfunction in practice once you computed it? How is it represented? Once you have it, how does it assign different values to each molecular structure?

**Questions:**

I would appreciate any time you can take to answer some of my questions.
1.  107 "for the  transfer operator we can only observe a noisy evaluation of the output": Do you mean a noisy version of the expectation that defines a transfer operator? Or that the output one can observe after simulating stochastic dynamics is "noisy"?
2. I assume your underlying goal is to discover CVs as the eigenfunctions of your learned generators. Your generators can be learned from biased enhanced sampling simulations where e.g. more transitions occur. Then we can obtain the CVs from your generator. With the CVs we can run biased simulations. Why are these biased simulations more informative than the biased simulations which you ran to train your generator? Can we compute different quantities from your CV biased simulations? Is it just a matter of your CVs being better biases than the bias of the original enhanced sampling bias?

**Limitations:**

The paper discusses some limitations such as not being time dependent - the significance of which I cannot assess. Limitations such as the limited evaluations and missing understanding of when the method fails seem more significant to me but are not mentioned or explained why they are present.

---

> ### Author Rebuttal · Authors · 2024-08-07
>
> We appreciate the reviewer's insightful evaluation and valuable comments. In what follows, we aim to address the highlighted weaknesses and respond to the reviewer's questions.
>
> ## Summary & strengths:
> - Thank you for your summary, while it captures the general idea of the paper, maybe it lacks one important aspect that, to the best of our knowledge, __we are the first to show__ that covariance reweighting in transfer operator (TO) models built from biased simulations fails, while infinitesimal generator (IG) models do not suffer from this effect. Indeed __we prove that one can learn eigenvalues and eigenfunctions__ of the IG via its resolvent __in a scalable and statistically consistent__ way from a single biased trajectory.
>
> - Related to the first strength, please refer to our general reply, part _Clarity of presentation_. Besides proposing Alg. 1 (methodological contribution), we also __theoretically prove its statistical consistency__. Let us provide the context: data-driven IG methods emerged relatively recently in the literature [1,15,20,21,46], and up to our knowledge, the only method that possesses finite-sample statistical guarantees for the proper spectral estimation is [21]. However, all of the above approaches analyzed learning from _unbiased_ simulations, which is unfeasible in many realistic scenarios. Moreover, the method in [21] suffers from scalability issues inherent to kernel methods. On the other hand, a deep learning method for the IG [46] was applied to biased simulations , but its statistical consistency was an open question.
>
> ## Weaknesses:
>
> 1. Noting that learning the spectral decomposition of IG is a problem of interest besides protein dynamics, e.g. crystallization, we agree with the suggested best practice. We did our best to meet such a requirement, as reported in _Experiments_ of our general reply and the attached pdf.
>   - We believe that the main reason why IG and TO methods are not initially tested on larger scale systems is the complexity of the task. Namely, recovering proper time scales and the transition state ensemble is a complex task that highly depends on the descriptors of the system and their effective dynamics. Indeed for large proteins one of the main challenges is to design appropriate descriptors so that effective Langevin dynamics can reveal the true dynamics of a protein [D]. These are highly dependent on the protein, and their design is a problem per se. Once this has been resolved, the next challenging task is to explore the rare transitions by biasing dynamics and recover proper time scales and transition state ensembles. While our method is designed to solve the latter, the former is an active field of research that currently limits the development of the full pipeline for large scale complex molecular systems.
>   - In order to fully address the question on the capability of our method, we have performed an additional experiment. To the best of our knowledge, the most complex benchmark for all TO and IG methods is the (very long) unbiased simulation dynamics of a small protein, typically Chignolin [7,19]. So, we designed a biased simulation experiment based on our method, which is, to the best of our knowledge, the first experiment of its kind for generator learning.
>   - Concerning baselines for ALDP, since we don’t have ground-truth it would be just a qualitative comparison.
> 2. Most non DL CV are heuristics based on chemical/physical intuition, they are system dependent and often suboptimal. This has stimulated intense research towards ML based alternatives. In this context, TO/IG methods are well-motivated, since eigenfunctions of these operators are “optimal” CVs for biasing simulations, see e.g. [A]. While some classical ML methods are available for this [21,23], they are intractable at large scale since at each time step one needs to compute the kernel function with respect to the current ensemble of the simulation.
> 3. Our method is as expensive as standard CV based methods, but it reliably leads to more accurate results, which is important when one wants to reconstruct the free energy profile and use them iteratively to learn the generator. See also our additional experiment on Chignolin, see the global reply and the attached pdf where we highlight the computational speedup relative to unbiased bruteforce simulations.
> 4. We incorporated your suggestion in the general reply on the general pipeline. Here we address your related questions. First, since the input of the DNN encoder in the representation learning part are the descriptors, the input dimension depends on the problem. On the other hand, the output (latent features) dimension is typically very small,  reflecting the number of slow time-scales to learn (typically around 3-5). Second, the choice of using separated DNNs per latent feature is made to aid learning of orthonormal features, since, depending on the architecture, shared weights may implicitly introduce prohibitive constraints. Third, eigenfunctions, defined in line 13 of Alg. 1, map descriptors of the system to a real number that, when properly scaled, has mean zero and variance 1 w.r.t. equilibrium distribution.
> 5. Eigenfunctions can be plotted in the plane of relevant features to analyze the data. For example, for alanine dipeptide we have for each configuration the values of the angles $\phi$ and $\psi$ and the values of the eigenfunction, therefore, we may plot the points in the $\phi-\psi$ plane colored according to the eigenfunction value.
>
> ## Questions:
> 1. If we understand well, two statements are equivalent. Indeed the noise of the feature is $z(X_{t+s})  - \mathbb{E}[z(X\_{t+s})\vert X_t] = z(X\_{t+s}) -[ {\cal T}\_s z ] (X_t)$, and from one trajectory we only observe an instance of the r. v. $z(X_{t+s})$, and not the output of TO.
> 2. Please see our general reply.
>
> Ref:
> [D] Zhang et al., Effective dynamics along given reaction coordinates and reaction rate theory. Faraday Discuss. (2016)

---

> > ### Comment · Reviewer_BQ7h · 2024-08-11
> >
> > Thank you for these nice explanations! I fully agree that studying and recovering dynamics from biased simulations is an important problem and appreciate any ML work done for it. I think the paper should be accepted.
> >
> >
> > I appreciate the chignolin experiment.
> > 1. Minor: Could you put your work a bit into context with "Implicit Transfer Operator Learning: Multiple Time-Resolution Surrogates for Molecular Dynamics" https://arxiv.org/abs/2305.18046 . They only learn the transfer operator. However, they also provide experiments on Chignolin. Are there inherent reasons why learning the generator might scale worse than learning the operator?
> >
> > My main concern (maybe it was naive) was in thinking that CVs (which you are learning by learning the generator and using its eigenfunctions) main or only value is in using them for further biased simulations. Thus, the question of why one biased simulation with your CVs would be better than the first biased simulation. This also seems valuable, and you suggest it in the general rebuttal, but do not try it. \
> > Important:\
> > However, as far as I understand, you suggest that I was wrong in assuming that learning the generator is only valuable for obtaining CVs and using them for biasing simulations. Learning them is also valuable to e.g. recover the free energy surface from a biased simulation or to recover dynamical quantities about transitions - please let me know if this understanding is correct.
> >
> > (sorry for the delayed reply - I will be quicker to respond in the remaining days)

---

> > > ### Author Response · Authors · 2024-08-12
> > > **Acknowledgement of reviewer's comments**
> > >
> > > We thank the reviewer for appreciating our rebuttal and suggesting our paper for acceptance. In the following we address the reviewer's additional questions.
> > >
> > > - __Concerning the reference.__ Thanks for bringing this paper to our attention, we will include it in the revision. In this work the authors build their model from D.E. Shaw dataset formed by a long trajectory containing already all the necessary information for training, without the need for biasing. They learn a transition kernel (i.e. a conditional probability to go from $X_t$ to $X_{t+\Delta t}$), which, as discussed in Sec. 3, has an inherent difficulty to be adapted to biased simulations.
> > >
> > > - __Scaling of generator algorithms.__ First, note that transfer operator (TO) based methods apply only to equally spaced data and that the sampling frequency  $1/\Delta t$ must be high enough to distinguish all relevant time-scales in the dynamics. Otherwise, since TO eigenvalues are $e^{\lambda_i \Delta t}$, small spectral gaps complicate learning (see Thm. 3 [23]). Conversely, our IG method, which uses gradient information, is time-scale independent, handles irregularly spaced measurements, and does not rely on time discretizations. This important aspect allows one to learn from biased simulations without notorious time-lag bottlenecks inherent to TO methods.  Alas, as there is no free lunch,  this incurs higher computational complexity. However, his complexity is to an extent mitigated through our representation learning, by exploiting automatic differentiation tools in deep learning.
> > >
> > > - __IG’s eigenfunctions.__ Indeed, the learned eigenfunctions of the IG can be used as CVs, which in fact are optimal CVs, as discussed in the general reply, see  [A]. Moreover, the leading eigenfunctions of IG encode the true dynamics. Hence, once they are properly learned, no more biasing is needed. For example, they can be used to infer the transition mechanism, see for instance figure 2, c) in the alanine dipeptide experiment, where we show that even with few transitions, we manage to recover a linear relationship between $\theta$ and $\phi$. Another important aspect is that one can build good approximations of all transfer operators $\cal{T}_{\Delta t}$, $\Delta t>0$, from the leading eigenpairs of IG, enabling forecasting of system’s observables see e.g. [21].
> > >
> > > If there are any remaining concerns and/or questions, we are happy to discuss more.

---

> ### Comment · Reviewer_BQ7h · 2024-08-12
>
> Thank you very much for confirming some points, and for addressing the reference. I would like to take the freedom to raise my score from 6 to 8 and to increase my confidence.

---

> > ### Author Response · Authors · 2024-08-12
> > **Acknowledgement to the reviewer**
> >
> > We are happy that our replies were helpful. We thank reviewer for all their comments and discussions, which I will improve our paper. We commit to incorporate them all in the revision.

---

### Official Review · Reviewer_txkX · 2024-07-12

**Soundness:** 3
**Presentation:** 3
**Contribution:** 3
**Rating:** 7
**Confidence:** 1

**Summary:**

In this paper, the authors investigate the possibility of estimating the leading eigenvalues and the corresponding eigenvectors for the evolution operators of Langevin dynamics using biased simulations. To this end, they rely on strong statistical guarantees and on the use of deep learning regression to build a suitable Hiltbert space that approximates the evolution operator of the unbiased process using only biased data. They evaluate the reliability of this approach in three increasingly complex problems and show that this approach successfully recovers the slowest relaxation modes in toy models and obtains better approximate values for the eigenvalues than competing state-of-the-art methods.

**Strengths:**

The work deals with a very important problem in chemistry and physics, namely the identification of the slowest modes of a dynamic process by means of simulations. In complex cases, simulations are not long enough to observe the most prominent transitions, and enhancing sampling methods are used to facilitate the observation of rare events. The problem is that it is often difficult to identify the key collective variables to facilitate the required slow movements. In this paper, the authors propose a simple way to do this using biased simulation data based on strong theoretical guarantees.

**Weaknesses:**

I find it difficult to read the paper and follow it, perhaps because I am too far from the field

**Questions:**

I did not understand well the physical meaning of the modes they extract with their approach. Could they give a physical explanation of them or of the order parameter associated?

**Limitations:**

Authors discuss correctly the limitations

---

> ### Author Rebuttal · Authors · 2024-08-07
>
> We appreciate the reviewer's insightful evaluation and valuable comments. In what follows, we aim to address the highlighted weaknesses and respond to the reviewer's questions.
>
> ## Weaknesses:
> We hope that our general reply brings more clarity. We commit to incorporate all the suggestions and additional discussions in the revised version of the paper. If the reviewer would like us to clarify any other specific aspects, we are happy to address them during the discussion period.
>
> ## Questions:
> Thank you for raising this question, in the following we try to briefly review the usefulness of generator eigenfunctions (“modes”) for interpretability and control of Langevin dynamics. We plan to add it in the revision to further clarify the rationale of our method for the general reader. From the spectral decomposition of the generator in Equation (4) and its link to the transfer operator in Equation (5) , it can be seen that each mode $f_i(x)\langle f_i,f \rangle$ (essentially the eigenfunction $f_i$) is related to a timescale of the process ($-\lambda_i>0$). This means that eigenfunctions corresponding to the slow time-scales (eigenvalues close to zero ) give information on __metastable states__. Recalling that normalized eigenfunctions (in $\mathcal{L}_\pi^2$) have mean zero and unit variance, they tend to have constant values for ensembles $x$ in such meta-stable states (meaning at time-scale $-\lambda_i>0$ there is no dynamics), while it changes sign when ensembles move from one meta-stable state to another. This is what the reviewer called “order parameter”. But it is more than this, because contrary to an order parameter which only discriminates two states, the eigenfunction will also give valuable information on how the transition takes place, see illustrative example in appendix E.1 and corresponding Figure 5. This feature of the eigenfunctions is of particular relevance, since the knowledge of rare transitions can be used by experimentalists to accelerate or slow down the process, see e.g. Wei Zhang, Christof Schütte, _Understanding recent deep‐learning techniques for identifying collective variables of molecular dynamics_. PAMM 2023, 23 (4).

---

> > ### Comment · Reviewer_txkX · 2024-08-09
> >
> > I really appreciated the authors' explanations, including the new figures (pipeline and the experiment with the proteins). I will change my rating to accept.

---

### Official Review · Reviewer_MSBv · 2024-07-13

**Soundness:** 3
**Presentation:** 3
**Contribution:** 3
**Rating:** 6
**Confidence:** 4

**Summary:**

This paper studies an infinitesimal generator approach for learning the eigenfunctions of evolution operators for Langevin SDEs. Due to the slow mixing caused by the high potential barriers, direct learning from simulation data can be sample inefficient. Biased simulation (based on a biased potential) is used to explore the space faster, and importance weights are constructed to get an unbiased loss function; such unbiased construction is more natural and feasible in the generator approach compared to the existing transfer operator approach. The minimizer of the resulting quadratic loss function, given a dictionary of basis functions, can be obtained by ridge regression type algorithms as it is a linear method. The authors also propose a loss function to learn a good dictionary that approximates the space of eigenfunctions more accurately; this nonlinear dictionary learning improves accuracy. Experiments on molecular dynamics benchmarks demonstrate the effectiveness of the approach.

**Strengths:**

The introduction and motivation are exceptionally well written and demonstrate the benefits of the generator approach compared to the transfer operator approach for biased simulations. The numerical experiments, especially in Figure 1, show significant improvements in accuracy compared to previous methods.

**Weaknesses:**

The mathematical presentation of the technical details, namely sections 4 and 5, could be made clearer. Many notations are introduced, but the ideas seem simple; see Questions. I would appreciate an overarching description highlighting the key idea and the difference between the proposed approach and existing works using generators.

**Questions:**

- Is the estimator in Section 4 simply the Galerkin approximation of the inverse of $\eta I - \mathcal{L}$ with the basis functions in $\mathcal{H}$?
I found the descriptions in Section 4 complicated and not easy to digest. As an example, for equation (11), by definition, the formula will differentiate the term $\|\chi_\eta(x) - \hat{G}^Tz(x)\|_2$ (which is not differentiable at zero) which looks strange. And I didn't understand the sentence "we contrast the resolvent ..." on page 5, line 182: what do the authors mean by "contrast" here? And the explanation of motivation of using the generator regression problem (11) rather than the mean square error also appears less clear.

- I am also curious about the mechanism behind the improvement of accuracy in Figure 2. It appears to me that the approach in this paper first uses the loss function in equation (19) to find approximate eigenfunctions and then uses the generator approach in section 4 to refine the estimation of the eigenfunctions and eigenvalues. If so, an understanding of which step of the two plays the significant role will be useful. For example, if the authors apply the generator approach in section 4 to the approximate eigenvalues obtained in the work of Zhang et al. (2022), will the resulting accuracy in Figure 2 also be significantly improved?

- Potential typos: Page 2, line 67: "principle" -> "principled".
Page 3, line 125-126: the notation of $C_{\gamma}$ is not introduced. Moreover, it seems $\lambda_i = \log \mu_i / t$ rather than $\log (\mu_i/t)$.
Page 8, line 282, "approqch"

- Page 3, line 129: "If t is chosen too small, the cross-covariance matrices will be too noisy for slowly mixing processes" Could the authors elaborate more on this?

**Limitations:**

Limitations are discussed.

---

> ### Author Rebuttal · Authors · 2024-08-07
>
> We appreciate the reviewer's insightful evaluation and valuable comments. In what follows, we aim to address the highlighted weaknesses and respond to the reviewer's questions.
>
> ## Weaknesses:
>
> Thank you for motivating us to present the overarching summary of our approach. While we discuss this in the general reply, we would like to provide a few more relevant details here.
> - First, data-driven generator methods emerged relatively recently in the literature [1,15,20,21,46], and up to our knowledge, the only method that possesses finite-sample statistical guarantees for the proper spectral estimation is [21]. However, all of the above approaches analyzed learning from unbiased simulations, which is unfeasible in many realistic scenarios. Moreover, the method in [21] suffers from scalability issues inherent to kernel methods. On the other hand, deep learning methods for the generator were applied to biased simulations [46], but their statistical consistency was not proven.  __In sharp contrast to all of the above, our work is the first to show that one can learn eigenvalues and eigenfunctions__ of the infinitesimal generator via its resolvent __in a scalable and statistically consistent way from biased simulations__.
> - To summarize, we extend the method in [21] by __incorporating biased dynamics__, developing appropriate __representation learning__, and designing the overall scalable and statistically consistent approach.  Our method is twofold: first, we learn a basis set using Theorem 2 (representation), and then we learn the generator’s resolvent on this basis set using Theorem 1 (regression). This approach is common in many fields, notably in transfer operator approaches [24,29], but is applied for the first time ever, to the best of our knowledge, to the case of the infinitesimal generator. In doing so, we followed the core idea of [21] to formulate the problem of learning the resolvent so that the geometry of the process is efficiently exploited via the energy norm, see also reply to Q1. Practically, this means that one can avoid inverting the shifted generator by rescaling the norm in which one learns the resolvent. This allows one to work in the latent space where the state becomes represented as $(\eta I - L)^{1/2} z$ and all the relevant inner products and norms can be computed using the features $z$ and their gradients.
>
> ## Questions:
>
> - __Q1:__ In fact, the estimator is not a Galerkin projection of the resolvent onto $\mathcal{H}$ as a subspace of $\mathcal{L}^2_\pi$. This would be the case if in (11) we replace the energy with the expectation w.r.t. invariant distribution. However, note that this is not feasible since we cannot compute/estimate the action of the resolvent. To solve this issue we change the domain of the problem from  $\cal{L}^2_\pi$ (expectation) to $\cal{H}\^\eta\_\pi$ (energy). So, the reviewer can think of our estimator as a __(regularized) Galerkin projection of the resolvent acting on $\cal{H}\^\eta\_\pi$ onto a subspace $\cal{H}$.__ The reason why we do so, as shown in (12), lies in the fact that Hilbert Schmidt norm of an operator $A$ on $\cal{H}^\\eta_\\pi$  becomes $\rm{tr}(A^*(\eta I-L)A)$. That is, the features $z\in\mathcal{L}^2_\pi$ are transformed into features $\tilde z:=(\eta I - L)^{1/2}) z$, so inner products $\langle \widetilde z, \widetilde z’ \rangle = \langle z,(\eta I - L) z’ \rangle$ and $\langle \widetilde z, (\eta I - L)^{-1} \widetilde z’ \rangle = \langle z, z’ \rangle$ can be now estimated from data. Using this trick we neatly avoid estimating the action of the resolvent, and no differentiation in (11) is needed since integral in $\chi_\eta$ is essentially canceled out by differential operator $\eta I -L$. This is what we meant by contrasting the resolvent in line 182. To further clarify our approach to the reader, we will include this discussion in the revision.
>
> - __Q2:__ Do we correctly understand that the reviewer is referring to Figure 1 instead of 2, since there we compare to [46] (Zhang et al. 2022)? If so, let us note that, as pointed out, we can use both methods for representation learning (loss (19) or loss 39 from [46]) to identify a subspace and then apply our regression method. Based on the theoretical background given in [46], we believe that their method indeed reduces representation error in Theorem 1, making this approach sound. However, when we performed this additional test, the first eigenvalue went from 0.043 to 0.013, the second one went from 1.13 to 4.19. This indicates that even with regression the result does not significantly improve. To strengthen our point, we also did the opposite operation: we computed the Rayleigh quotient from our features without doing any regression, just as is done in the work [46] we obtain for the first eigenvalue 1.79e-4 and 0.36 for the second one, which are close to the ones found by regression and very close to the ground truth values. We are speculating that the main reason is in the efficiency of our loss and its robustness on the hyperparameter choices.
>
> - __Q3 and Q4:__ Thank you for pointing out typos, we will correct them. Also, sorry for the confusion, we intended to say that if $t$ is chosen too small w.r.t. the mixing time, the noise in the corresponding transfer operator regression dominates the signal, meaning that the learning becomes harder. We will correct this discussion in the revision.

---

> ### Comment · Reviewer_MSBv · 2024-08-09
>
> Thank you for the detailed response. I have some follow-up points.
>
> - **Q1**. I now agree that the estimator is a regularized Galerkin projection of the resolvent onto the space of basis functions, using the energy inner product. This interpretation seems more convenient from an expository standpoint. It would be helpful if the authors can incorporate this perspective into the paper and clarify the writing. In my experience, the current mathematical exposition, with its various operators, norms and formulas, did not provide a very pleasant reading experience.
>
> - **Q2**. Yes, it is Figure 1. Thank you. The explanation and results are interesting. Do the authors have any insights regarding the superiority of the loss (19) for representation learning? It seems to me the loss is not that simple and easy to optimize or compute. There is a penalization term to make the $z_i$ normalized and one has the hyper parameter $\alpha$ to tune. Furthermore, based on the authors' response, do I understand correctly that the primary improvement stems from the loss function (19) for representation learning, which is more significant than the regression formulation proposed in the paper? If so I think these insights should be added to the paper.

---

> > ### Author Response · Authors · 2024-08-09
> > **Further discussion**
> >
> > We thank the reviewer for the prompt reply.
> > - __Q1__ We agree that Galerkin projection interpretation seems more convenient from an expository standpoint. So, based on the reviewer’s feedback, we propose to modify the content in lines 185-191, by briefly explaining why Galerkin projection should be done in energy space $\cal{H}^\eta_\pi$ instead of $\cal{L}^2_\pi$ space. Then, since the statistical learning risk formulation is the key to understand how to generalizes w.r.t. (unobserved) true distribution $\pi$ from biased simulations, we briefly discuss its equivalence with the Galerkin viewpoint. We believe that this change can help the reader to better grasp the main idea behind the energy, and connect it to the material that follows. Does the reviewer find this proposal adequate?
> >
> > - __Q2: about the loss:__ To answer this question, let us compare (19) to the two losses used in the most related works, [24] and [46]. While the former one (only tested on unbiased dynamics of a toy system) depends on one hyperparameter, the latter loss has $1+m$ hyperparameters, where $m$ is the latent dimension. Importantly, __both losses suffer from statistically biased estimation__ of gradients when optimized over the sample distributions, which may negatively impact the optimization.  Moreover, as reported in [46] (see also lines 287-289) additional $m$ hyperparameters in loss of [46] are delicate to tune. __In sharp contrast__, our empirical loss, as stated in Theorem 2, __is an unbiased estimate__ of the true loss in Equation (18), and we didn’t experience any particular difficulty in tuning our only hyperparameter $\alpha>0$. Note also that, in principle $\alpha>0$ is optional to speed up the training, and we may as well use $\alpha=0$. Concerning the computation, we respectfully disagree that the loss is hard to compute. Indeed, for two size $n$ batches of data, it just relies on computation of covariances of features $C$ and their gradients $W$, which is of the order $\cal{O}(n m^2 d)$. Since in the molecular dynamics setting $m$ is typically small, computing the loss is very efficient, similarly to losses in self-supervised learning based on canonical correlations, see e.g. [A]. The only computational bottleneck lies in using the gradients for $W$. However, we believe that this is a necessary price to pay to be able to generalize properly from biased simulations, as motivated in Sec. 3. Finally, we remark that optimization of our loss follows interpretable dynamics, see Figure 3 of the Appendix where plateaus reveal discovery of new relevant features, helping practitioners to decide when to stop the training.
> >
> > - __Q2: about the contribution:__ As the reviewer rightfully recognizes, introducing loss (19) for representation learning is an important contribution of this work. However, our main focus is on __biased simulations__, or, in other words,  __learning dynamics that ML algorithm did not observe__. With this in mind, our main contributions are summarized in lines 67-74. For the reviewer's convenience we rephrase and slightly expand them here. To the best of our knowledge, for the first time
> > - __(1)__ we __formalize the problem of learning__ the spectral decomposition of the infinitesimal generator __from biased simulations__ as a regression,
> > - __(2)__ we __derive empirical estimators with guaranteed statistical consistency__ (Theorem 1), and
> > - __(3)__ we propose an end-to-end __representation learning + regression__ pipeline that efficiently scales to large problems.
> > - __(4)__ we __empirically validate our methodology__ on molecular dynamics datasets of increasing complexity.  That said, __we will follow reviewer’s suggestion and emphasize better the important role of representation learning__ for the generator regression.
> >
> > Once more we thank the reviewer, and if any other doubt remains or clarification is needed, we are happy to discuss in more detail.
> >
> > Ref.
> >
> > [A] Balestriero et al., A Cookbook of Self-Supervised Learning. Arxiv:2304.12210 (2023).

---

> > > ### Comment · Reviewer_MSBv · 2024-08-13
> > > **Acknowledgement of rebuttal by authors**
> > >
> > > Thank you for the detailed response. They are very clear.

---

### Author Rebuttal · Authors · 2024-08-07

We wish to thank all reviewers for their insightful evaluation of our paper. We appreciate all their comments and remarks, which we’ll incorporate in our revision. Before addressing each review in detail, we’d like to point out some general remarks that apply to all of them.

##   Clarity of presentation
To improve the clarity of our work in Sec. 4 and 5, and emphasize its impact, let us provide a general overarching view of our method. Dealing with rare events implies a large timescale, thus generator’s eigenvalues close to zero, whose eigenfunctions reveal transitions between meta-stable states. To solve these learning tasks via transfer operator approaches one typically needs a prohibitively large amount of data from simulations with very small time-increments. We overcome this problem by proposing to work with the infinitesimal generator which is, contrary to transfer operators, time-scale independent and, as we show, can be successfully combined with biased simulations.
In particular, we propose to work with the resolvent of the infinitesimal generator, which shares the same eigenfunctions with the generator. However, this approach introduces a new issue: an operator needs to be inverted. To overcome this, we work in a new Hilbert space $\cal{H}\_\pi^\eta$ with inner product defined in Eq. (10). In this space, the Hilbert Schmidt norm of an operator $A$ on $\cal{L}^2_\pi$ becomes $\rm{tr}(A^*(\eta I-L)A)$, that is, features $z$ are transformed into features $(\eta I-L)^{1/2}) z$. Using this trick, __the resolvent may be regressed easily__ in a theoretically grounded way knowing only the diffusion part of $L$, see also [21] for more details on this approach using kernels. On the other hand, to obtain a method that is truly scalable to large problems, one needs a good dictionary of functions on which to regress the generator. This  __representation learning problem__ is addressed in Section 5, where we use the same weighted norm to define a novel  loss. While we show in Thm. 2 that deep neural network representations, when properly trained, converge towards the eigenfunctions of the resolvent, due to imperfect dataset or parameter choices, this convergence might be slow. However, the learned features are useful to identify a subspace of the $\cal{L}^2_\pi$ domain with small representation error, and one can efficiently regress the resolvent with the method presented in Sec. 4 to obtain the overall statistical consistency from Thm. 1.

## General pipeline
Following the comments of the referees, we also believe that a description of our method pipeline would be beneficial. We propose it in the form of Figure 1 in the attached pdf, and commit to include it in the revision, together with the following discussion and more detailed version of the current pseudocode (Alg. 1) in the appendix:
First, one chooses a molecular system to study, and, based on this choice, identifies a __collective variable (CV)__ on which to bias the dynamics to observe transitions from one state to the other. Often this CV is heuristic and suboptimal, leading to only a few transitions. To apply our method (and any deep-learning based method), one then needs to __choose descriptors__ of the system to put as input of the model. These descriptors will encode the symmetries of the system (often global translation/rotation) and only take into account the most important degrees of freedom: for example for the folding process of a protein in water, one may only use the interatomic distances between heavy atoms. This phase is highly system dependent, and the complexity of choosing descriptors grows with the system size. Once descriptors are chosen, a __representation is learned__ and the derivatives present in the loss function are computed with respect to atomic positions. After learning the representation, __the resolvent is regressed__ on the learned dictionary of functions. If the initial dataset does not have enough transitions, it might be that the eigenfunctions obtained are not good enough. In that case, one can use the learned eigenfunction as a collective variable for biasing the simulation to collect more transitions and further improve our model of the generator, and hence the final estimation of its eigenfunctions. Note, however, that the last point is based on the fact that the eigenfunctions of the generator are optimal CVs, see  [A]. While we did not need to implement it in our experiments, future work will study more complex systems where this iterative process can be beneficial.

## New experiment
Finally, to showcase the power of our method, we implement it for the Chignolin miniprotein: we first performed a 1 microsecond biased simulation using as CV the deep-TDA one [B]. This allowed us to gather transitions. Then, this trajectory was used to train our method, which is, to the best of our knowledge, the largest scale work on generator learning. To validate our results, we trained our method on a very long unbiased trajectory of 106 microseconds from D.E. Shaw research [C]. The results are presented in Fig. 2 of the attached pdf. The two presented eigenfunctions are completely similar, but one is from only 1 microsecond long simulation, while the other is from 106 microseconds (so a 106 times more costly computations). This  shows that __our work on biased simulations paves the way towards scalable and reliable estimation of dynamical quantities in realistic problems of molecular dynamics__.

We once more thank all the reviewers whose comments inspired the discussion above which, we believe, further strengthens our paper, and demonstrates its impact.
The Authors

### Ref.
[A] Zhang & Schütte, Understanding recent deep‐learning techniques for identifying collective variables of molecular dynamics. PAMM (2023)

[B]  Trizio & Parrinello, From enhanced sampling to reaction profiles. J. Phy. Ch. Letters (2021)

[C] Lindorff-Larsen et al. “How Fast-Folding Proteins Fold”. Science (2011)

---

### Decision · Program_Chairs · 2024-09-25

**Decision:**

Accept (poster)

**Comment:**

This paper introduces a biased potential to improve the efficiency of sampling and inferences of simulated SDE. Reviewers agree that the paper is tackling an important and difficult problem, especially in modeling protein dynamics. They introduce a new method supported by theory and experiments. If accepted, please make sure to include new experimental results and incorporate improvements that stemmed from the rebuttal discussion.